# CORRELATED ATTENTION IN TRANSFORMERS FOR MULTIVARIATE TIME SERIES

## ABSTRACT

Multivariate time series (MTS) analysis prevails in real-world applications such as finance, climate science and healthcare. The various self-attention mechanisms, the backbone of the state-of-the-art Transformer-based models, efficiently discover the temporal dependencies, yet cannot well capture the intricate cross-correlation between different features of MTS data, which inherently stems from complex dynamical systems in practice. To this end, we propose a novel correlated attention mechanism, which not only efficiently captures feature-wise dependencies, but can also be seamlessly integrated within the encoder blocks of existing well-known Transformers to gain efficiency improvement. In particular, correlated attention operates across feature channels to compute cross-covariance matrices between queries and keys with different lag values, and selectively aggregate representations at the sub-series level. This architecture facilitates automated discovery and representation learning of not only instantaneous but also lagged cross-correlations, while inherently capturing time series auto-correlation. When combined with prevalent Transformer baselines, correlated attention mechanism constitutes a better alternative for encoder-only architectures, which are suitable for a wide range of tasks including imputation, anomaly detection and classification. Extensive experiments on the aforementioned tasks consistently underscore the advantages of correlated attention mechanism in enhancing base Transformer models, and demonstrate our state-of-the-art results in imputation, anomaly detection and classification.

## 1 INTRODUCTION

Multivariate time series (MTS) are time series encompassing multiple dimensions for capturing different features of the original data, where each dimension corresponds to a univariate time series. MTS analysis is ubiquitous in real-world applications such as imputation of missing data in geoscience (López et al., 2021), anomaly detection of monitoring data in aeronautics (Hundman et al., 2018b), classification of hearbeat data for fetal assessment (Kampouraki et al., 2009), and weather prediction (Wu et al., 2022b). Thanks to its immense practical value, there has been increasing interest in MTS analysis (Wen et al., 2023; Wu et al., 2023; Lim & Zohren, 2021; Zhang & Yan, 2023).

The recent advancement of deep learning has facilitated the development of many models with superior performance (Li et al., 2021b; Wu et al., 2023). Specifically, the large class of Transformer-based models (Wen et al., 2023; Wu et al., 2022b; Zhang & Yan, 2023; Zhou et al., 2022; Liu et al., 2022; Vaswani et al., 2017; Du et al., 2023b) is the most prominent and has demonstrated great potential for their well-known capability to model both short-range and long-range temporal dependencies (Wen et al., 2023). In addition to temporal dependencies, feature-wise dependencies, which are cross-correlation between the variates of MTS, are central to MTS analysis (Cao et al., 2020) and studied in the deep learning literature via convolution neural network (CNN) (Lai et al., 2018) or graph neural network (GNN) (Wu et al., 2020; Cao et al., 2020). Nevertheless, for existing Transformer-based models (e.g. (Li et al., 2019; Zhou et al., 2021; Wu et al., 2022b)), the embedding method is insufficient for capturing such cross-correlation between different variates of MTS (Zhang & Yan, 2023), which motivated the authors therein to propose CrossFormer as the first Transformer explicitly utilizing feature-wise dependencies for MTS forecasting. Despite its promising performance, CrossFormer deploys a convoluted architecture, which is isolated from other prevalent Transformers with their own established merits in temporal modelling and specifically designed for only MTS forecasting,

thereby lacking flexibility. Consequently, it remains under-explored whether modelling feature-wise dependencies could also improve Transformer-based models' performances in other non-predictive tasks, which cover a wide range of real-world applications and include prominently imputation, anomaly detection and classification. Moreover, all the previous work (Wu et al., 2020; Cao et al., 2020; Zhang & Yan, 2023) on capturing feature-wise dependencies in MTS analysis are limited in scope to forecasting, rely on ad-hoc mechanisms in their rigid pipelines, and thus do not fully leverage the capability to model temporal dependencies of existing powerful Transformers. Motivated by the nascent literature of the aforementioned problems and the success of Transformer-based models in MTS analysis, we raise the following central question of this paper:

*How can we seamlessly elevate the broad class of existing and future Transformer-based architectures to also capture feature-wise dependencies? Can modelling feature-wise dependencies improve Transformers' performance on non-predictive tasks?*

We affirmatively answer this question by proposing a novel correlated attention mechanism that efficiently learns the cross-correlation between different variates of MTS and can be seamlessly integrated with the encoder-only architecture of well-known Transformers, thereby being applicable to a wide range of non-predictive tasks. In addition to the conventional cross-correlation, the correlated attention captures simultaneously auto-correlation, the backbone of Autoformer (Wu et al., 2022b), and lagged cross-correlation. Lagged cross-correlation has been inherently critical in MTS data (John & Ferbinteanu, 2021; Chandereng & Gitter, 2020), yet vastly ignored by the literature of Transformer-based models. For raw MTS data of production planning (e.g. (Contreras-Reyes & Idrovo-Aguirre, 2020)) as an example, it may take some lagged interval for the increase in the demand rate to be reflected in the production rate. Instead of the usual temporal dimension, correlated attention operates across feature channels to compute cross-covariance matrices of between queries and keys with different lag values, and further select the pairs with highest correlations for aggregating representations at the sub-series level. For seamless integration with the encoder block of base Tranformers such as (Vaswani et al., 2017; Liu et al., 2022) with their respective temporal attentions, the original multi-head attention is modified to include the heads using both the temporal attentions from the base model and our correlated attentions. This design directly augments the embedded layer of the base Transformer with cross-correlation information in its representation learning. Experimentally, correlated attention, when plugged into prevalent Transformer baselines, consistently boosts the performance of the base models and results in state-of-the-art benchmark for Transformer-models in various tasks . The contributions of the paper can be summarized as follows:

- We propose a novel correlated attention mechanism that efficiently learns both the instantaneous and lagged cross-correlations between different variates of MTS, as well as auto-correlation of series. To the best of our knowledge, this is the first work that presents a Transformer architecture that aims to explicitly learn the lagged cross-correlation.

- Correlated attention is flexible and efficient, where it can be can be seamlessly plugged into encoder-only architectures of well-known Transformers such as (Vaswani et al., 2017; Liu et al., 2022) to enhance the performance of the base models. It naturally augments the embedded layer of base Transformers, having been known vastly for temporal modelling (Zhang & Yan, 2023), with feature-wise dependencies. Furthermore, the modularity of correlated attention will permit its adoption in and benefit future Transformer architectures.

- Extensive experiments on imputation, anomaly detection and classification demonstrate that correlated attention consistently improves the performance of base Transformers and results state-of-the-art architectures for the aforementioned tasks.

## 2 RELATED WORK

**Multivariate Time Series Analysis.** The surge of advanced sensors and data stream infrastructures has lead to the tremendous proliferation of MTS data (Wen et al., 2022; Esling & Agon, 2012). In response, MTS analysis, which spans a multitude of tasks including but not limiting to imputation (Du et al., 2023b), anomaly detection (Blázquez-García et al., 2020), classification (Fawaz et al., 2019) and forecasting (Lim & Zohren, 2021), has been increasingly crucial. In recent years, many deep learning models have been proposed for MTS analysis and achieved competitive performance (Lai et al., 2018; Franceschi et al., 2020; Wen et al., 2023; Gu et al., 2022). Specifically, multilayer perceptron

(MLP) methods (Oreshkin et al., 2020; Challu et al., 2022) adopt MLP blocks for modelling temporal dependencies. Temporal Convolutional Networks (TCNs) (Lea et al., 2016; Franceschi et al., 2020) leverage CNN or recurrent neural network (RNN) along the temporal dimension to capture temporal dependencies. RNN-based models (Hochreiter & Schmidhuber, 1997; Lai et al., 2018) use state transitions and recurrent structure to model temporal variations. In order to capture cross-correlation, recent work (Yu et al., 2018; Cao et al., 2020; Wu et al., 2020) deploy GNNs to directly model cross-dimension dependencies. Nevertheless, these neural networks rely on RNN and CNN to model temporal dynamics, which are are known to be inefficient in capturing long-range temporal dependencies (Zhang & Yan, 2023). TimesNet (Wu et al., 2023) models temporal 2D-variation for both intraperiod and interperiod variations via residual structure TimesBlock.

**Transformers in MTS Analysis.** Originating from natural language processing (NLP) domain, Transformers (Vaswani et al., 2017) have shown great success when adapted to MTS analysis (Zhou et al., 2022; Li et al., 2019; Zhou et al., 2021; Liu et al., 2022; Wu et al., 2022b; Du et al., 2023b) thanks to their capability to capture both short-range and long-range temporal dependencies (Wen et al., 2023). Recently, Liu et al. (2022) performed series stationarization to attenuate time series non-stationarity. Wu et al. (2022b) proposed Autoformer with decomposition architecture and auto-correlation mechanism for better modelling of long-range temporal dependencies. Crossformer (Zhang & Yan, 2023) uses dimension-segment-wise embedding and a hierarchical architecture to better learn both the cross-time and cross-dimension dependencies.

**Modelling Cross-correlation in Time Series.** Capturing feature-wise dependencies in MTS analysis has been a long lasting problem, where such cross-correlation in MTS data stems from natural processes (Li et al., 2021a) and complex cyper-physical systems (CPSs) (Wu et al., 2021; Cirstea et al., 2018). Accurate forecasting of correlated MTS can reveal the underlying dynamics of the system including trend and intrinsic behavior (Yang et al., 2013a), and detect outliers (Kieu et al., 2018). To capture the MTS correlation, previous work have proposed the adoptions of hidden Markov models (Yang et al., 2013b) and spatio-temporal (ST) graphs (Cirstea et al., 2021) as the modeling primitives, specialized neural network architectures for correlated MTS forecasting (Wu et al., 2021; Cirstea et al., 2018), and methods based on cross-correlation analysis (Yuan et al., 2016; Kristoufek, 2014). Nevertheless, most of these approaches focused on either forecasting with ST correlation, which arises from the proximity of the MTS sensors' locations and is only applicable to CPSs, or ad-hoc MTS analysis. (Lai et al., 2018) models long and short term temporal patterns with deep neural networks in MTS forecasting. Crossformer (Zhang & Yan, 2023) was the first Transformer-based architecture that explicitly utilizes both temporal and feature-wise dependencies for MTS forecasting. Yet, for non-predictive tasks such as imputation, anomaly detection and classification, there has been no Transformer with specialized modelling of feature-wise dependencies. Moreover, while lagged cross-correlation is inherent in MTS data, for which various statistical tools (John & Ferbinteanu, 2021; Chandereng & Gitter, 2020; Probst et al., 2012; Shen, 2015) have been developed for testing and analysis, time series Transformers in the literature have not leveraged this information in their mechanisms to improve performance of target applications.

## 3 METHODOLOGY

In this Section, we first review the two representative well-known temporal attention mechanisms, namely the self-attention (Vaswani et al., 2017) and de-stationary attention (Liu et al., 2022), and the multi-head attention architecture commonly used in a wide range of Transformer-based models such as (Vaswani et al., 2017; Liu et al., 2022; Du et al., 2023a; Zhou et al., 2021; Wu et al., 2022b) and more. Next, we discuss the current limitation of conventional temporal attentions in modelling feature-wise dependencies. This then motivates us to propose the correlated attention mechanism, which operates across the feature channels for learning cross-correlation among variates, and combine it with existing temporal attentions in the mixture-of-head attention architecture to improve the performance of the base Transformers.

### 3.1 BACKGROUND

**Self-attention.** Self-attention, first proposed in the vanilla Transformer (Vaswani et al., 2017), operates on the query, key and value matrices. In particular, given the input the matrix $X \in \mathbb{R}^{T \times d}$, where $T$ is the sequence length and $d$ is feature dimension of the model, the model linearly projects

$X$ into queries, keys and values respectively as $Q = XW^Q, K = XW^K$ and $V = XW^V$, where $W^Q \in \mathbb{R}^{d \times d_k}, W^K \in \mathbb{R}^{d \times d_k}$ and $W^V \in \mathbb{R}^{d \times d_v}$ are parameter matrices. Taking queries $Q$, keys $K$ and values $V$ as input, the self-attention returns the output matrix as follows:

$$\text{SELF-ATTENTION}(Q, K, V) = \text{SOFTMAX}\left(\frac{1}{\sqrt{d_k}} QK^\top\right)V. \tag{1}$$

The computational complexity of self-attention is $O(d_k T^2)$ due to pairwise interactions along the time dimension $T$.

**De-stationary Attention.** To handle non-stationary real-world MTS data, Non-stationary Transformer (Liu et al., 2022) performs series stationarization for better predictability and adopts the de-stationary attention mechanism to alleviate the over-stationarization and recover the intrinsic information into temporal dependencies. Specifically, after the normalization module, Non-stationary Transformer operates over the stationarized series $X' = (X - \mathbf{1}\mu_X^\top)/\sigma_X$ with the mean vector $\mu_X$ and covariance $\sigma_X$, and obtain the stationarized queries, keys and values respectively as $Q' = (K - \mathbf{1}\mu_Q^\top)/\sigma_X$, $K' = (K - \mathbf{1}\mu_K^\top)/\sigma_X$ and $V' = (V - \mathbf{1}\mu_V^\top)/\sigma_X$ with the mean vectors $\mu_Q, \mu_K$ and $\mu_V$. Then, it can be proven that (Liu et al., 2022):

$$\text{SOFTMAX}\left(\frac{1}{\sqrt{d_k}} QK^\top\right) = \text{SOFTMAX}\left(\frac{1}{\sqrt{d_k}}(\sigma_X^2 Q'K'^\top + \mathbf{1}\mu_Q^\top K^\top)\right),$$

which motivates their design of de-stationary attention utilizing multilayer perceptron (MLP) layer to directly learn the positive scaling scalar $\xi \approx \sigma_X^2$ and shifting vector $\Delta \approx K\mu_Q$, and returning the output matrix:

$$\text{DE-STATIONARY-ATTENTION}(Q', K', V') = \text{SOFTMAX}\left(\frac{1}{\sqrt{d_k}}(\xi Q'K'^\top + \mathbf{1}\Delta^\top)\right)V'. \tag{2}$$

The computational complexity of de-stationary attention is $O(d_k T^2)$ without accounting for the MLP module.

While there have been a multitude of other temporal attention mechanisms (e.g. (Zhou et al., 2021; Du et al., 2023b; Zhou et al., 2022)) that usually follow ad-hoc design for specific tasks, the two representative attention mechanisms above are the backbones of some of most primitive Transformers that have robust and competitive performances on a variety of tasks. Next, we present the multi-head attention module, which adopts the temporal attention as its component and commonly used in a wide range of Transformer-based models (e.g. (Vaswani et al., 2017; Liu et al., 2022; Du et al., 2023a; Zhou et al., 2021)).

**Multi-head Attention.** Multi-head attention, proposed along with self-attention in the vanilla Transformer (Vaswani et al., 2017), combines multiple temporal attentions to jointly attend to information from different representation subspaces. In particular, it concatenates $h$ heads, where each head is the output from some temporal attention and $h$ is a hyperparameter, and then performs linear projection for the final output. Formally, multi-head attention is written as follows:

$$\text{MULTI-HEAD-ATTENTION}(X) = \text{CONCAT}(head_1, head_2, ..., head_h)W^O$$
$$\text{where } head_i = \text{TEMPORAL-ATTENTION}(XW_i^Q, XW_i^K, XW_i^V). \tag{3}$$

In the Equation 3, $W^O \in \mathbb{R}^{hd_v \times d}$ is parameter matrix and TEMPORAL-ATTENTION can take the form of any mechanism, such as the two aforementioned self-attention and de-stationary attention, or any other in the literature (Vaswani et al., 2017; Liu et al., 2022; Du et al., 2023a; Zhou et al., 2021).

## 3.2 CORRELATED ATTENTION BLOCK AND MIXTURE-OF-HEAD ATTENTION

In this Section, we first take a deeper look at how the design of self-attention (or more generally temporal attention) can limit its capability of modeling feature-wise dependencies, while approaches in the literature of Transformers' attention design may be insufficient to capture the cross-correlation in MTS. This motivates us to propose the correlated attention block (CAB) to efficiently learn the feature-wise dependencies and can be seamlessly plugged into ubiquitous encoder-only Transformer architectures for performance improvement. Next, we demonstrate how the computation for CAB can be further accelerated via Fast Fourier Transform (FFT) thanks to the Cross-correlation Theorem.

### 3.2.1 LIMITATION OF TEMPORAL ATTENTION

One interpretation for the powerful temporal modeling capacity of Transformers is that, with the queries $Q = [\mathbf{q}_1, \mathbf{q}_2, ..., \mathbf{q}_T]^\top$ and keys $K = [\mathbf{k}_1, \mathbf{k}_2, ..., \mathbf{k}_T]^\top$ expressed in time-wise dimension, the matrix $QK^\top \in \mathbb{R}^{T \times T}$ in the computation of self-attention (Equation 1) contains pairwise inner-products $\mathbf{q}_i^\top \mathbf{k}_j$ of time-dimension vectors, and thus intuitively resembles the notion of correlation matrix between different time points of MTS data. Nevertheless, feature-wise information, where each of the $d_k$ features corresponds to an entry of $\mathbf{q}_i \in \mathbb{R}^{d_k \times 1}$ or $\mathbf{k}_j \in \mathbb{R}^{d_k \times 1}$, is absorbed into such inner-product representation; this thus makes self-attention unable to explicitly leverage the feature-wise information in its representation learning. In the context of computer vision, El-Nouby et al. (2021) considered a cross-covariance attention mechanism that instead computes $\hat{K}^\top \hat{Q} \in \mathbb{R}^{d_k \times d_k}$, where $\hat{K}$ and $\hat{Q}$ are $\ell_2$-normalized versions of $K$ and $Q$, as the cross-covariance matrix along the feature dimension. However, while this simple design is suitable for capturing instantaneous cross-correlation in static image applications as considered therein, it is insufficient to capture the cross-correlation of MTS data which is coupled with the intrinsic temporal dependencies. In particular, the variates of MTS data can be correlated with each other, yet with a lag interval– this phenomenon is referred to as lagged cross-correlation in MTS analysis (John & Ferbinteanu, 2021; Chandereng & Gitter, 2020; Probst et al., 2012; Shen, 2015). Additionally, a variate in MTS data can even be correlated with the delayed copy of itself, the phenomenon of which is termed auto-correlation. Wu et al. (2022b) proposed Autoformer with the auto-correlation mechanism, but their rigid framework is specifically designed for and achieves competitive performance in long-term forecasting. Given the nascent literature of modules to augment a broad class of powerful Transformers with yet less-efficient modelling capabilities of cross-correlation and auto-correlation, we hereby aim to derive a flexible and efficient correlated attention mechanism that can elevate existing Transformer-based models.

### 3.2.2 CORRELATED ATTENTION BLOCK

We proceed to present our correlated attention block (CAB), which is comprised of three consecutive components: normalization (Equation 4), lagged cross-correlation filtering (Equation 5), and score aggregation (Equation 6).

**Normalization.** In the normalization step, we perform column-wise $\ell_2$ normalization of $Q$ and $K$, respectively resulting in $\hat{Q}$ and $\hat{K}$ as:

$$\hat{Q} = \text{NORMALIZE}(Q), \quad \hat{K} = \text{NORMALIZE}(K), . \tag{4}$$

**Lagged Cross-correlation Filtering.** We first present the overview of the lagged cross-correlation filtering step as follows:

$$l_1, l_2, ..., l_k = \underset{l \in [1, T-1]}{\arg\text{TopK}} \left\{ \lambda \cdot \text{DIAGONAL}\big(\text{ROLL}(\hat{K}, l)^\top \hat{Q}\big) \right.$$
$$\left. + (1 - \lambda) \cdot \text{NON-DIAGONAL}\big(\text{ROLL}(\hat{K}, l)^\top \hat{Q}\big) \right\}, \tag{5}$$

where $\lambda \in [0, 1]$ is a learnable parameter and $\arg\text{TopK}(.)$ is used to select the $k = c\lceil \log(T) \rceil$ (with $c$ being a hyperparameter) time lags which incur the highest cross-correlation scores to be described in more details now. The purpose of the previous normalization step is to unify the feature-wise variates into the same scale, so that $\text{ROLL}(\hat{K}, l)^\top \hat{Q}$ can better serve as a notion of cross-correlation matrix in feature-wise dimension between that queries $\hat{Q}$ and the lagged keys $\text{ROLL}(\hat{K}, l)$. Here, for $X \in \mathbb{R}^{T \times d_k}$, the $\text{ROLL}(X, l)$ operation shifts the elements of $X$ vertically, i.e. along the time-dimension, during which entries shifted over the first position are then re-introduced at the last position. This rolling operation helps generating lagged series representation. In order to formally define our lagged cross-correlation filtering step (Equation 5), we hereby consider the two operations $\text{DIAGONAL}(.)$ and $\text{NON-DIAGONAL}(.)$ on square matrix that respectively sum up the absolute values

of diagonal entries and non-diagonal entries. Specifically, given a matrix $A \in \mathbb{R}^{d_k \times d_k}$, we then have:

$$\text{DIAGONAL}(A) = \sum_{i=1}^{d_k} |A_{ii}|,$$

$$\text{NON-DIAGONAL}(A) = \sum_{i,j \in [1,d_k]: i \neq j} |A_{ij}|.$$

Recall from stochastic process theory (Chatfield, 2004; Papoulis, 1965) that for any real discrete-time process $\{\mathcal{X}_t\}$, its auto-correlation $R_{\mathcal{X},\mathcal{X}}(l)$ can be computed by $R_{\mathcal{X},\mathcal{X}}(l) = \lim_{L \to \infty} \frac{1}{L} \sum_{t=1}^{L} \mathcal{X}_t \mathcal{X}_{t-l}$. With the normalized queries $\hat{Q} = [\hat{\mathbf{q}}_1, \hat{\mathbf{q}}_2, ..., \hat{\mathbf{q}}_{d_k}]$ and normalized keys $\hat{K} = [\hat{\mathbf{k}}_1, \hat{\mathbf{k}}_2, ..., \hat{\mathbf{k}}_{d_k}]$ expressed in feature-wise dimension where $\hat{\mathbf{q}}_i, \hat{\mathbf{k}}_j \in \mathbb{R}^{T \times 1}$, any $i^{th}$ diagonal entry of $\text{ROLL}(\hat{K}, l)^\top \hat{Q}$ takes the form $\left(\text{ROLL}(\hat{K}, l)^\top \hat{Q}\right)_{ii} = R_{\hat{\mathbf{q}}_i, \hat{\mathbf{k}}_i}(l) = \sum_{t=1}^{T} (\hat{\mathbf{q}}_i)_t \cdot (\hat{\mathbf{k}}_i)_{t-l}$ and thus can serve as an approximation (with multiplicative factor) for the auto-correlation of variate $i$. This idea was also harnessed in the design of auto-correlation attention (Wu et al., 2022b). Consequently, given a lag $l$, the quantity $\text{DIAGONAL}\left(\text{ROLL}(\hat{K}, l)^\top \hat{Q}\right)$, which aggregates over the absolute values of all diagonal entries, scores the total auto-correlation of all the feature variates, while the quantity $\text{NON-DIAGONAL}\left(\text{ROLL}(\hat{K}, l)^\top \hat{Q}\right)$ scores the the total cross-correlation between different pairs of feature variates. The final cross-correlation score incurred by time lag $l$ is then the weighted (convex) combination of $\text{DIAGONAL}\left(\text{ROLL}(\hat{K}, l)^\top \hat{Q}\right)$ and $\text{NON-DIAGONAL}\left(\text{ROLL}(\hat{K}, l)^\top \hat{Q}\right)$ with a learnable weight $\lambda$ as shown in Equation 5. For high-dimensional MTS data where not all pairs of variates are highly correlated and/or auto-correlation is the more significant factor, the learnable parameter $\lambda$ helps automatically untangle such relations and balance the representation learning between auto-correlation and cross-correlation of interacting features. Then $k = c\lceil \log(T) \rceil$ (with $c$ being a hyperparameter) time lags $l_1, l_2, ..., l_k$, which get the highest cross-correlation scores, are selected through the TopK operation to be used in the next step.

**Score Aggregation.** Finally, the CAB performs sub-series aggregation for the final output via:

$$\text{CORRELATED-ATTENTION}(Q, V, K) = (1 - \beta) \cdot \text{ROLL}(V, 0) \cdot \text{SOFTMAX}\left(\frac{1}{\tau} \text{ROLL}(\hat{K}, 0)^\top \hat{Q}\right)$$

$$+ \beta \cdot \sum_{i=1}^{k} \text{ROLL}(V, l_i) \cdot \text{SOFTMAX}\left(\frac{1}{\tau} \text{ROLL}(\hat{K}, l_i)^\top \hat{Q}\right), \quad (6)$$

where $\beta \in [0, 1]$ and $\tau > 0$ are learnable parameters. In particular, for every chosen lag $l_i$, we also roll the values matrix $V$ by $l_i$ to align similar sub-series with the same phase position. Then, each $\text{ROLL}(V, l_i) \cdot \text{SOFTMAX}\left(\frac{1}{\tau} \text{ROLL}(\hat{K}, l_i)^\top \hat{Q}\right)$ is a convex combination in feature dimension (as opposed to time dimension in self-attention in Equation 1) of the corresponding token embedding in the delayed values $\text{ROLL}(V, l_i)$. The final score aggregation in Equation 6 is the weighted (convex) combination of the "instantaneous" score $\text{ROLL}(V, 0) \cdot \text{SOFTMAX}\left(\frac{1}{\tau} \text{ROLL}(\hat{K}, 0)^\top \hat{Q}\right)$ and the "lagged" total score $\sum_{i=1}^{k} \text{ROLL}(V, l_i) \cdot \text{SOFTMAX}\left(\frac{1}{\tau} \text{ROLL}(\hat{K}, l_i)^\top \hat{Q}\right)$ with a learnable weight $\beta$.

**Efficient computation of CAB.** In its current form, the computation complexity of CAB is $O(d_k^2 T^2)$. Specifically, for every lag $l$, the computation of $\text{ROLL}(\hat{K}, l)^\top \hat{Q}$ takes $O(d_k^2 T)$ time. With our choice of $k = O(\log(T))$, Equation 6 takes $O(d_k^2 T \log(T))$ time. Nevertheless, since Equation 5 requires iterating over all $T - 1$ lags $l \in [1, T - 1]$, each of which costs $O(d_k^2 T)$, the total complexity is $O(d_k^2 T^2)$. We hereby present how to alleviate the computation in Equation 5 via FFT, thereby resulting in the accelerated complexity of $O(d_k^2 T \log(T))$. This is enabled via the Cross-correlation Theorem (Lahiri, 2016), which, given two finite discrete time series $\{\mathcal{X}_t\}$ and $\{\mathcal{Y}_t\}$, permits the sliding inner product $(\mathcal{X} \star \mathcal{Y})(l) = \sum_{t=1}^{T} \mathcal{X}_{t-l} \mathcal{Y}_t$ of different lag values $l \in [0, T-1]$ being computed efficiently via FFT as:

$$\mathcal{S}_{\mathcal{XY}}(f) = \mathcal{F}(\mathcal{X}_t)\mathcal{F}^*(\mathcal{Y}_t) = \int_{-\infty}^{+\infty} \mathcal{X}_t e^{-i2\pi t f} dt \overline{\int_{-\infty}^{+\infty} \mathcal{Y}_t e^{-i2\pi t f} dt}$$

$$(\mathcal{X} \star \mathcal{Y})(l) = \mathcal{F}^{-1}(\mathcal{S}_{\mathcal{XY}}(f)) = \int_{-\infty}^{+\infty} \mathcal{S}_{\mathcal{XY}}(f) e^{i2\pi f l} df, \quad (7)$$

for $l \in [0, T-1]$, where $\mathcal{F}$ and $\mathcal{F}^{-1}$ are FFT and FFT inverse, and $*$ is the conjugate operation. Particularly, given $\bar{K}, \bar{Q} \in \mathbb{R}^{T \times d_k}$, we first compute $\mathcal{F}(\bar{K}), \mathcal{F}(\bar{Q}) \in \mathbb{R}^{(T/2+1) \times d_k}$ in the frequency domain. Let $\mathcal{F}(.)_i$ be the $i^{th}$ column of these FFTs. We then compute $\mathcal{F}(\bar{K})_i \mathcal{F}^*(\bar{Q})_j$ for all $i, j \in [1, d_k]$. Finally, the inverse FFTs of these products would give $\mathcal{F}^{-1}\big(\mathcal{F}(\bar{K})_i \mathcal{F}^*(\bar{Q})_j\big) = \big[\big(\text{ROLL}(\bar{K}, 0)^\top \bar{Q}\big)_{ij}, \big(\text{ROLL}(\bar{K}, 1)^\top \bar{Q}\big)_{ij}, ..., \big(\text{ROLL}(\bar{K}, T-1)^\top \bar{Q}\big)_{ij}\big]$ for $i, j \in [1, d_k]$. Thus, we can gather data to obtain $\text{ROLL}(\bar{K}, l)^T \bar{Q}$ for all $l \in [0, T-1]$. As each of FFT and inverse FFT takes $O(T \log(T))$, CAB achieves the $O(d_k^2 T \log(T))$ complexity. We note that the cross-correlation computation required by CAB is more complicated and strictly subsumes auto-correlation and the invoked Cross-correlation Theorem is more generalized version of the Wiener–Khinchin Theorem, as used by (Wu et al., 2022b) for auto-correlation computation.

**Differences Compared to Autoformer.** Since the CAB aims to capture the lagged cross-correlation, which is relevant to yet more generalized than the auto-correlation module in Autoformer, we believe it is crucial to emphasize the main differences. First, Autoformer overall is a decomposed encoder-decoder architecture proposed for long-term forecasting, so its auto-correlation module is specifically designed to work with series seasonalities extracted from various series decomposition steps of Autoformer. On the other hand, CAB ensures flexibility with any input series representation by deploying normalization step and learnable temperature coefficient $\lambda$ reweighting the correlation matrices. Second, while Autoformer computes purely auto-correlation scores and aggregates their exact values for TopK, CAB computes cross-correlation matrices and aggregates the absolute values of such entries for TopK in Equation 5 (as correlation can stem from either positive or negative correlation). Finally, to facilitate robustness to different input series representation, CAB adopts learnable weights $\lambda$ in TopK operation, which balances between auto-correlation and cross-correlation, and $\beta$ in sub-series aggregation, which balances between instantaneous and lagged cross-correlation.

### 3.2.3 Mixture-of-head Attention

For seamless integration of CAB with a broad class of encoder-only Transformer architectures using multi-head attention component (e.g. (Vaswani et al., 2017; Liu et al., 2022; Du et al., 2023a; Zhou et al., 2021)), we propose mixture-of-head attention that leverages a mixture of both temporal attentions and correlated attentions. mixture-of-head attention modifies multi-head attention (Equation 3) to also incorporate CAB as follows:

$$\text{MIXTURE-OF-HEAD-ATTENTION}(X) = \text{CONCAT}(head_1, head_2, ..., head_h)W^O$$

$$\text{where } head_i = \begin{cases} \text{TEMPORAL-ATTENTION}(XW_i^Q, XW_i^K, XW_i^V), \text{ if } i \leq m \\ \text{CORRELATED-ATTENTION}(XW_i^Q, XW_i^K, XW_i^V), \text{ otherwise} \end{cases}, \qquad (8)$$

where $m$ is a threshold hyperparameter that controls the split between temporal attention and correlated attention. This uncomplicated modification to the base architecture of multi-head attention allows CAB to be flexibly plugged into a wide range of existing and future Transformers.

## 4 Experiments

As CAB is a plug-in attention for encoder-only Transformer architectures, we extensively experiment on three mainstream MTS non-predictive tasks including imputation, anomaly detection and classification on real-world datasets. Ablation studies are provided in Appendix B. While focusing on non-predictive tasks, we provide preliminary results on MTS long-term forecasting in Appendix C. Run-time analysis is presented in Appendix D.

Table 1: Dataset Summary

| MTS Analysis Tasks | Benchmarking Datasets | Metrics | Sequence Length |
|---|---|---|---|
| Imputation | ETTm1, ETTm2, ETTh1, ETTh2, Electricity, Weather | MSE, MAE | 96 |
| Anomaly Detection | SMD, MSL, SMAP, SWaT, PSM | Precision, Recall, F1-score (%) | 100 |
| Classification | UEA (10 subsets) | Accuracy (%) | 29-1751 |

**Experiment Benchmarks.** Following (Zhou et al., 2021; Wu et al., 2023; Zerveas et al., 2021), we extensively benchmark over the following real-world datasets: ETTh1 and ETTh2 (Electricity

Transformer Temperature-hourly) (Zhou et al., 2021), ETTm1 and ETTm2 (Electricity Transformer Temperature-minutely) (Zhou et al., 2021), Electricity (Trindade, 2015), Weather (Wetterstation), SMD (Su et al., 2019), MSL (Hundman et al., 2018a), SMAP (Hundman et al., 2018a), SWaT (Mathur & Tippenhauer, 2016), PSM (Abdulaal et al., 2021) and UEA Time Series Classification Archive (Bagnall et al., 2018). A summary of the datasets for benchmark is given in Table 1.

**Baselines.** We compare with *TimesNet* (Wu et al., 2023)[1], the current state-of-the-art deep learning model on these three tasks (though not being Transformer-based), *DLinear* (Zeng et al., 2022), and the prevalent Transformer-based models including vanilla *Transformer* (Vaswani et al., 2017), *Non-stationary Transformer* (Liu et al., 2022), which has been shown to consistently achieve competitive results on a variety of tasks, *FEDformer* (Zhou et al., 2022), and *Autoformer* (Wu et al., 2022b). In fact, Nonstationary Transformer and FEDformer are the state-of-the-art Transformer-models for respectively imputation and anomaly detection in the recent benchmarks (Wu et al., 2023). For classification, we also consider *Flowformer* (Wu et al., 2022a), the state-of-the-art Transformer-model.

**Our Models.** We integrate CAB (through the mixture-of-head attention) into two representative models: *Transformer* (Vaswani et al., 2017) and *Nonstationary Transformer* (Liu et al., 2022).

## 4.1 IMPUTATION

**Setup.** Due to uncertainties of natural processes and malfunction of sensors, missing data is common in MTS, thereby hindering direct adoption of off-the-shelf models. MTS imputation has thus gathered much research interest (López et al., 2021). To exemplify real-world scenario commonly facing data missing problem, we consider six datasets from electricity and weather domain for benchmark: ETTh1 and ETTh2 (ETT-hourly) (Zhou et al., 2021), ETTm1 and ETTm2 (ETT-minutely) (Zhou et al., 2021), Electricity (Trindade, 2015) and Weather (Wetterstation). Each dataset is split into three sets of training set, validation set, and test set respectively with ratio $60\%, 20\%$ and $20\%$. Time-series data is generated by selecting every 96 consecutive steps as a sample. To test the models under different missing data rate, we randomly mask the time points with the ratio of $\{12.5\%, 25\%, 37.5\%, 50\%\}$. We adopt the mean square error (MSE) and mean absolute error (MAE) as the metrics.

**Results.** The results are depicted in Table 2. Nonstationary+CAB and Transformer+CAB improve over Nonstationary and Transformer in respectively five and four datasets out of the total of six datasets. Nonstationary+CAB achieves state-of-the-art results surpassing TimesNet on five datasets.

Table 2: Imputation task over six datasets. The missing data rate is $\{12.5\%, 25\%, 37.5\%, 50\%\}$ and series length is 96. We highlight the best results and the second best results.

| Datasets | Mask Ratio | TimesNet (Wu et al., 2023) | | Nonstationary (Liu et al., 2022) | | Nonstationary+CAB (Ours) | | Transformer (Vaswani et al., 2017) | | Transformer+CAB (Ours) | | FEDformer (Zhou et al., 2022) | | DLinear (Zeng et al., 2022) | | Autoformer (Wu et al., 2022b) | |
|---|---|---|---|---|---|---|---|---|---|---|---|---|---|---|---|---|---|
| | | MSE | MAE | MSE | MAE | MSE | MAE | MSE | MAE | MSE | MAE | MSE | MAE | MSE | MAE | MSE | MAE |
| ETTm1 | 12.5 % | 0.019 | 0.092 | 0.026 | 0.107 | 0.018 | 0.087 | 0.023 | 0.105 | 0.022 | 0.104 | 0.035 | 0.135 | 0.058 | 0.162 | 0.034 | 0.124 |
| ETTm1 | 25 % | 0.023 | 0.101 | 0.032 | 0.119 | 0.02 | 0.097 | 0.030 | 0.121 | 0.031 | 0.123 | 0.052 | 0.166 | 0.080 | 0.193 | 0.046 | 0.144 |
| ETTm1 | 37.5 % | 0.029 | 0.111 | 0.039 | 0.131 | 0.030 | 0.112 | 0.037 | 0.135 | 0.039 | 0.140 | 0.069 | 0.191 | 0.103 | 0.219 | 0.057 | 0.161 |
| ETTm1 | 50 % | 0.036 | 0.124 | 0.047 | 0.145 | 0.037 | 0.125 | 0.045 | 0.148 | 0.050 | 0.157 | 0.089 | 0.218 | 0.132 | 0.248 | 0.067 | 0.174 |
| **Average** | | 0.027 | 0.106 | 0.036 | 0.126 | 0.026 | 0.105 | 0.034 | 0.127 | 0.036 | 0.131 | 0.062 | 0.177 | 0.093 | 0.206 | 0.051 | 0.150 |
| ETTm2 | 12.5 % | 0.018 | 0.080 | 0.021 | 0.088 | 0.016 | 0.076 | 0.125 | 0.264 | 0.136 | 0.271 | 0.056 | 0.159 | 0.062 | 0.166 | 0.023 | 0.092 |
| ETTm2 | 25 % | 0.020 | 0.085 | 0.024 | 0.096 | 0.018 | 0.082 | 0.195 | 0.323 | 0.152 | 0.288 | 0.080 | 0.195 | 0.085 | 0.196 | 0.026 | 0.10 |
| ETTm2 | 37.5 % | 0.023 | 0.091 | 0.027 | 0.103 | 0.024 | 0.092 | 0.217 | 0.343 | 0.179 | 0.312 | 0.110 | 0.231 | 0.106 | 0.222 | 0.030 | 0.108 |
| ETTm2 | 50 % | 0.026 | 0.098 | 0.030 | 0.108 | 0.027 | 0.099 | 0.257 | 0.378 | 0.211 | 0.340 | 0.156 | 0.276 | 0.131 | 0.247 | 0.035 | 0.119 |
| **Average** | | 0.022 | 0.088 | 0.026 | 0.099 | 0.021 | 0.087 | 0.199 | 0.327 | 0.170 | 0.303 | 0.101 | 0.215 | 0.096 | 0.208 | 0.029 | 0.105 |
| ETTh1 | 12.5 % | 0.057 | 0.159 | 0.060 | 0.165 | 0.047 | 0.148 | 0.063 | 0.178 | 0.070 | 0.189 | 0.070 | 0.190 | 0.151 | 0.267 | 0.074 | 0.182 |
| ETTh1 | 25 % | 0.069 | 0.178 | 0.080 | 0.189 | 0.064 | 0.171 | 0.089 | 0.212 | 0.098 | 0.223 | 0.106 | 0.236 | 0.180 | 0.292 | 0.090 | 0.203 |
| ETTh1 | 37.5 % | 0.084 | 0.196 | 0.102 | 0.212 | 0.085 | 0.195 | 0.115 | 0.242 | 0.137 | 0.264 | 0.124 | 0.258 | 0.215 | 0.318 | 0.109 | 0.222 |
| ETTh1 | 50 % | 0.102 | 0.215 | 0.133 | 0.240 | 0.106 | 0.216 | 0.140 | 0.270 | 0.162 | 0.286 | 0.165 | 0.299 | 0.257 | 0.347 | 0.137 | 0.248 |
| **Average** | | 0.078 | 0.187 | 0.094 | 0.201 | 0.076 | 0.182 | 0.102 | 0.226 | 0.117 | 0.241 | 0.117 | 0.246 | 0.201 | 0.306 | 0.103 | 0.214 |
| ETTh2 | 12.5 % | 0.040 | 0.130 | 0.042 | 0.133 | 0.039 | 0.129 | 0.205 | 0.329 | 0.212 | 0.354 | 0.095 | 0.212 | 0.100 | 0.216 | 0.044 | 0.138 |
| ETTh2 | 25 % | 0.046 | 0.141 | 0.049 | 0.147 | 0.044 | 0.139 | 0.283 | 0.397 | 0.228 | 0.355 | 0.137 | 0.258 | 0.127 | 0.247 | 0.050 | 0.149 |
| ETTh2 | 37.5 % | 0.052 | 0.151 | 0.056 | 0.158 | 0.051 | 0.150 | 0.285 | 0.392 | 0.265 | 0.378 | 0.187 | 0.304 | 0.158 | 0.276 | 0.060 | 0.163 |
| ETTh2 | 50 % | 0.060 | 0.162 | 0.065 | 0.170 | 0.059 | 0.160 | 0.327 | 0.418 | 0.319 | 0.415 | 0.232 | 0.341 | 0.183 | 0.299 | 0.068 | 0.173 |
| **Average** | | 0.049 | 0.146 | 0.053 | 0.152 | 0.048 | 0.145 | 0.275 | 0.384 | 0.256 | 0.376 | 0.163 | 0.279 | 0.142 | 0.259 | 0.055 | 0.156 |
| Electricity | 12.5 % | 0.085 | 0.202 | 0.093 | 0.210 | 0.081 | 0.198 | 0.148 | 0.276 | 0.143 | 0.269 | 0.107 | 0.237 | 0.092 | 0.214 | 0.089 | 0.210 |
| Electricity | 25 % | 0.089 | 0.206 | 0.097 | 0.214 | 0.087 | 0.204 | 0.161 | 0.285 | 0.165 | 0.283 | 0.120 | 0.251 | 0.118 | 0.247 | 0.096 | 0.220 |
| Electricity | 37.5 % | 0.094 | 0.213 | 0.102 | 0.220 | 0.093 | 0.209 | 0.170 | 0.292 | 0.168 | 0.290 | 0.136 | 0.266 | 0.144 | 0.276 | 0.104 | 0.229 |
| Electricity | 50 % | 0.100 | 0.221 | 0.108 | 0.228 | 0.098 | 0.215 | 0.177 | 0.296 | 0.173 | 0.295 | 0.158 | 0.284 | 0.175 | 0.305 | 0.113 | 0.239 |
| **Average** | | 0.092 | 0.210 | 0.100 | 0.218 | 0.089 | 0.207 | 0.164 | 0.287 | 0.162 | 0.284 | 0.130 | 0.259 | 0.132 | 0.260 | 0.101 | 0.225 |
| Weather | 12.5 % | 0.025 | 0.045 | 0.027 | 0.051 | 0.026 | 0.050 | 0.034 | 0.090 | 0.033 | 0.082 | 0.041 | 0.107 | 0.039 | 0.084 | 0.026 | 0.047 |
| Weather | 25 % | 0.029 | 0.052 | 0.029 | 0.056 | 0.029 | 0.056 | 0.036 | 0.089 | 0.034 | 0.085 | 0.064 | 0.163 | 0.048 | 0.103 | 0.030 | 0.054 |
| Weather | 37.5 % | 0.031 | 0.057 | 0.033 | 0.062 | 0.034 | 0.064 | 0.038 | 0.091 | 0.038 | 0.089 | 0.107 | 0.229 | 0.057 | 0.117 | 0.032 | 0.060 |
| Weather | 50 % | 0.034 | 0.062 | 0.037 | 0.068 | 0.041 | 0.074 | 0.042 | 0.095 | 0.046 | 0.105 | 0.183 | 0.312 | 0.066 | 0.134 | 0.037 | 0.067 |
| **Average** | | 0.030 | 0.054 | 0.032 | 0.059 | 0.032 | 0.061 | 0.038 | 0.091 | 0.038 | 0.090 | 0.099 | 0.203 | 0.052 | 0.110 | 0.031 | 0.057 |

---

[1]While results of TimesNet on forecasting and imputation are reproducible, we cannot recover its state-of-the-art results, from their released code, on anomaly detection and classification. We report here the results on such two tasks obtained from their released implementation and note that the relative ranking of baselines remains the same as in TimesNet benchmark (Wu et al., 2023), i.e. TimesNet is the best among the previous baselines.

## 4.2 ANOMALY DETECTION

**Setup.** Anomalies are inherent in large-scale data and can be caused by noisy measurements. We consider the five datasets vastly used for anomaly-detection benchmarks: SMD (Su et al., 2019), MSL (Hundman et al., 2018a), SMAP (Hundman et al., 2018a), SWaT (Mathur & Tippenhauer, 2016) and PSM (Abdulaal et al., 2021). We then follow (Xu et al., 2022; Shen et al., 2020) for pre-processing data that generates a set of sub-series via non-overlapped sliding window, and set the series length to 100. The original datasets SMD, MSL, SMAP, SWaT and PSM are splitted into collections of training set, validation set and test set following (Xu et al., 2022, Appendix K). We adopt Precision, Recall and F1-score (all in %) as the metrics, where higher values correspond to better performance.

**Results.** From Table 3, our model Nonstationary+CAB achieves the best average F1-score, surpassing TimesNet. Furthermore, CAB consistently and significantly improves the precision and F1-score, which is the more favorable metrics for balancing precision and recall, of the base Transformers.

Table 3: Anomaly detection task over five datasets. We report the Precision (P), Recall (R) and F1-score (F1)- the harmonic mean of precision and recall, and highlight the best results and the second best results.

| Datasets | TimesNet (Wu et al., 2023) | | | Transformer (Vaswani et al., 2017) | | | Transformer+CAB (Ours) | | | Nonstationary (Liu et al., 2022) | | | Nonstationary+CAB (Ours) | | | FEDformer (Zhou et al., 2022) | | | DLinear (Zeng et al., 2022) | | | Autoformer (Wu et al., 2022b) | | |
|---|---|---|---|---|---|---|---|---|---|---|---|---|---|---|---|---|---|---|---|---|---|---|---|---|
| | P | R | F1 | P | R | F1 | P | R | F1 | P | R | F1 | P | R | F1 | P | R | F1 | P | R | F1 | P | R | F1 |
| SMD | 87.88 | 81.54 | 84.59 | 83.58 | 76.13 | 79.56 | 78.36 | 65.25 | 71.20 | 88.33 | 81.21 | 84.62 | 90.43 | 82.33 | 86.19 | 87.95 | 82.39 | 85.08 | 83.62 | 71.52 | 77.10 | 88.06 | 82.35 | 85.11 |
| MSL | 89.55 | 75.29 | 81.80 | 71.57 | 87.37 | 78.68 | 89.70 | 73.66 | 80.90 | 68.55 | 89.14 | 77.50 | 88.02 | 72.83 | 79.71 | 77.14 | 80.07 | 78.57 | 84.34 | 85.42 | 84.88 | 77.27 | 80.92 | 79.05 |
| SMAP | 90.05 | 56.54 | 69.46 | 89.37 | 57.12 | 69.70 | 90.86 | 61.87 | 73.79 | 89.37 | 59.02 | 71.09 | 90.27 | 57.3 | 70.10 | 90.47 | 58.10 | 70.76 | 92.32 | 55.41 | 69.26 | 90.40 | 58.62 | 71.12 |
| SWaT | 90.95 | 95.42 | 93.13 | 68.84 | 96.53 | 80.37 | 99.67 | 68.89 | 81.47 | 68.03 | 96.75 | 79.88 | 90.55 | 95.41 | 92.92 | 90.17 | 96.42 | 93.19 | 80.91 | 95.30 | 87.52 | 89.85 | 95.81 | 92.74 |
| PSM | 98.51 | 96.29 | 97.39 | 62.75 | 96.56 | 76.07 | 99.34 | 82.92 | 90.39 | 97.82 | 96.76 | 97.29 | 98.25 | 96.13 | 97.58 | 97.31 | 97.16 | 97.23 | 98.28 | 89.26 | 93.55 | 99.08 | 88.15 | 93.29 |
| Average | 91.39 | 81.02 | 85.27 | 75.22 | 82.74 | 76.88 | 91.59 | 70.52 | 79.55 | 82.42 | 84.06 | 82.08 | 91.50 | 80.80 | 85.30 | 88.61 | 82.83 | 84.97 | 87.89 | 79.38 | 82.46 | 88.93 | 81.17 | 84.262 |

## 4.3 CLASSIFICATION

**Setup.** We select ten datasets from the UEA Time Series Classification Archive (Bagnall et al., 2018) following (Wu et al., 2023). These cover health care, audio recognition, transportation and other practical applications. The datasets are pre-processed similarly to (Zerveas et al., 2021, Appendix A) that assigns different series length for different subsets. We adopt the accuracy (%) as the metrics.

**Results.** As shown in Table 4, our model Transformer+CAB achieves the best overall result surpassing TimesNet. Moreover, CAB demonstrates consistent performance improvement when combined with either Transformer or Nonstationary Transformer.

Table 4: Classification task task over 10 datasets from UEA. The accuracies (%) are reported. We highlight the best results and the second best results.

| Datasets | TimesNet (Wu et al., 2023) | Transformer (Vaswani et al., 2017) | Transformer+CAB (Ours) | Nonstationary (Liu et al., 2022) | Nonstationary+CAB (Ours) | FEDformer (Zhou et al., 2022) | DLinear (Zeng et al., 2022) | Flowformer (Wu et al., 2022a) | Autoformer (Wu et al., 2022b) |
|---|---|---|---|---|---|---|---|---|---|
| Ethanol | 28.14 | 26.24 | 31.94 | 25.10 | 25.10 | 28.90 | 27.00 | 33.08 | 27.38 |
| FaceDetection | 67.31 | 67.93 | 71.11 | 68.70 | 69.40 | 68.55 | 67.25 | 67.08 | 54.63 |
| Handwriting | 29.88 | 29.53 | 29.06 | 31.41 | 30.12 | 18.47 | 18.94 | 27.18 | 13.18 |
| Heartbeat | 74.15 | 75.12 | 75.12 | 72.20 | 72.20 | 75.12 | 70.73 | 72.68 | 69.76 |
| JapaneseVowels | 97.57 | 97.03 | 97.84 | 96.22 | 95.68 | 96.76 | 94.86 | 98.65 | 94.86 |
| PEMS-SF | 89.02 | 78.03 | 86.71 | 82.66 | 75.14 | 86.71 | 80.35 | 86.71 | 82.66 |
| SCP1 | 91.13 | 91.13 | 91.47 | 83.28 | 82.94 | 57.00 | 88.05 | 89.08 | 59.39 |
| SCP2 | 52.78 | 53.89 | 56.11 | 50.00 | 55.55 | 49.44 | 52.78 | 54.44 | 53.89 |
| SpokenArabic | 98.68 | 98.45 | 99.05 | 98.82 | 98.91 | 98.32 | 96.54 | 98.95 | 98.82 |
| UWaveGesture | 86.88 | 86.25 | 85.94 | 81.56 | 85.94 | 44.06 | 81.25 | 86.88 | 45.63 |
| Average | 71.49 | 70.36 | 72.44 | 69.00 | 69.10 | 62.33 | 67.78 | 71.47 | 60.02 |

## 5 CONCLUSION AND FUTURE WORK

In this paper, we proposed the novel correlated attention block (CAB) that can efficiently learn the cross-correlation between variates of MTS data, and be seamlessly plugged into existing Transformer-based models for performance improvement. The modularity of CAB, which could be flexibly plugged into follow-up Transformer-architectures for efficiency gain, and the methodology behind our design of CAB, which is the first attention mechanism that aims to capture lagged cross-correlation in the literature, will greatly benefit future work on time series Transformers. Extensive experiments on imputation, anomaly detection and classification demonstrate the benefits of CAB for improving base Transformers, and result in state-of-the-art models for respective tasks. For future work, we will extend the design of CAB to be integrated into encoder-decoder Transformer-architectures for improving performance in MTS predictive tasks.

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
