## A    IMPLEMENTATION DETAILS

The models are implemented in PyTorch and experiments are run on a NVIDIA-SMI v100 GPU with 100Gb of storage. The learnable parameters are initialized as $\tau = 1$, and $\lambda = \beta = 1/2$. For MTS datasets with dimension $d < 70$, we set $d_{model} = d_k = 64$; otherwise, we use $d_{model} = d_k = 128$. For PEMS-SF, we set $d_{model} = d_k = 180$. If $d_k < 100$, we fix $\lambda = 1/2$ in Equation 5. The hyperparameter $c$ in the $k = c\lceil \log(T) \rceil$ of TopK is chose among $c \in \{1, 2, 3\}$, where we pick out the value with best result. For the mixture-of-head attention, we use $h = 16$ and $m = 8$, i.e. 8 heads for temporal attentions and 8 heads for correlated attentions. The loss for classification is entropy, while the loss for imputation and anomaly detection is MSE. We use ADAM for training with the default hyperparameter configuration. Batch size is set to 16 for imputation and classification, and 128 for anomaly detection. The number of epochs is set to 30. If the validation loss does not decreases for 10 epochs, the training is stopped.

## B    ABLATION STUDIES

The main components of CAB are the lagged cross-correlation filtering step (Equation 5), which involves the learnable $\lambda$ to untangle auto-correlation and cross-feature correlation, and score aggregation step (Equation 6), which involves the learnable $\beta$ to balance instantaneous cross-correlation and lagged cross-correlation. A key component for the seamless integration of CAB into base Transformers is the mixture-of-head attention (MOHA) that utilizes a mixture of temporal and correlated attention heads. To test each of the aforementioned components, we use the same experimental setting for classification as in Section 4, and consider the following ablation versions of Transformer+CAB, the best performing model for this task:

- **pure-CAB-Transformer**: *(testing MOHA)* In this model, we simply replace the self-attention of vanilla Transformer with the most basic correlated attention setting where lagged cross-correlation filtering step is disabled, i.e. no learning of $\lambda$, and $\beta$ is set to 0; hence score aggregation steps just returns the instantaneous cross-correlation. In short, self-attention is now replaced by:

$$\text{CORRELATED-ATTENTION}(Q, K, V) = V \text{SOFTMAX}(\frac{1}{\tau} \hat{K}^\top \hat{Q}),$$

  to test how this simple mechanism can take place of self-attention.

- **static-Transformer+CAB**: *(testing lagged cross-correlation filtering)* In this model, we use 8 self-attention heads and 8 correlated attention heads, and enable back the lagged cross-correlation filtering, yet hard-fix $\lambda = \beta = 1/2$, i.e. there is no learning of these parameters. This is to test the simplified lagged cross-correlation filtering's efficiency.

- **$\lambda$-Transformer+CAB**: *(testing $\lambda$)* This is the same as static-Transformer+CAB except that we now allow $\lambda$ to be learnable parameter. $\beta$ is fixed to $1/2$.

- **$\beta$-Transformer+CAB**: *(testing $\beta$)* This is the same as static-Transformer+CAB except that we now allow $\beta$ to be learnable parameter. $\lambda$ is fixed to $1/2$.

Table 5: Ablation studies where the original model Transformer+CAB is compared to the variants with different disabled components.

| Dataset/Method | Transformer+CAB | pure-CAB-Transformer | static-Transformer+CAB | $\lambda$-Transformer+CAB | $\beta$-Transformer+CAB |
|---|---|---|---|---|---|
| Ethanol | 31.94 | 30.79 | 28.14 | 29.28 | 30.42 |
| FaceDetection | 71.11 | 68.81 | 69.67 | 70.20 | 70.83 |
| Handwriting | 29.06 | 20.94 | 23.88 | 24.35 | 29.06 |
| Heartbeat | 75.12 | 72.68 | 75.61 | 74.15 | 75.12 |
| JapaneseVowels | 97.84 | 96.76 | 95.68 | 95.68 | 97.84 |
| PEMS-SF | 86.71 | 72.83 | 79.77 | 76.88 | 84.39 |
| SCP1 | 91.47 | 86.01 | 90.44 | 88.74 | 91.13 |
| SCP2 | 56.11 | 58.89 | 57.22 | 55.00 | 55.00 |
| SpokenArabic | 99.05 | 98.45 | 98.54 | 99.27 | 99.05 |
| UWaveGesture | 85.94 | 80.00 | 83.13 | 82.50 | 85.31 |
| **Average** | 72.44 | 68.62 | 70.01 | 69.60 | 71.82 |

**Results.** The results from Table 5 indicate the importance of every main component of the overall design of CAB. First, while MOHA is disabled, the pure-CAB-Transformer obtains poor performance, as opposed to the other three ablation variants. Second, despite being in its simplified version, lagged cross-correlation is crucial and demonstrates significant improvement from pure-Transformer+CAB to static-Transformer+CAB. The decrease in performance of $\lambda$-Transformer+CAB from static-Transformer+CAB demonstrates that for low-dimensional MTS data, leanrable $\lambda$ is unnecessary. Nevertheless, for high-dimensional data, such as FaceDetection, learnable $\lambda$ results in efficiency gain. Finally, from the good performance of $\beta$-Transformer+CAB, we conclude that the score aggregation step with learnable $\beta$ is important in the pipeline of CAB. All in all, for all the ablation versions, the drops in accuracies are insignificant, thereby showing the robustness of our model. Furthermore, the two most crucial components of the CAB that account for the most significant efficiency boost are the lagged cross-correlation filtering (as shown in static-Transformer+CAB versus pure-CAB-Transformer) and the learnable $\beta$ for balancing between instantaneous and lagged cross-correlation in Equation 6 (as shown in $\beta$-Transformer+CAB versus static-Transformer+CAB).

## C  ADDITIONAL EXPERIMENTAL RESULTS FOR LONG-TERM FORECASTING

The decoder architecture of many prevalent Transformers (e.g. (Vaswani et al., 2017; Liu et al., 2022)) is comprised of a masked multi-head attention block and a usual multi-head attention block (i.e. without masking). In its current form, CAB has not been designed to be fully integrated into decoder architecture of Transformers yet, since it lacks the suitable masking mechanism. Nevertheless, we still consider the naive design of decoder architecture that still maintains the masked multi-head attention block of the base Transformer, yet (for the non-masked block) deploys mixture-of-head attention block combining the base temporal attention with CAB. We then test the effectiveness of the above decoder architecture integrated with CAB in MTS long-term forecasting, and believe that with proper masking mechanism for CAB in the future work, the performance increase can be further improved. Specifically, we augment Nonstationary Transformer (Liu et al., 2022), well-known for its competitive performance in long-term forecasting, with CAB, and experiment Nonstationary+CAB on the two common datasets ETTh (Zhou et al., 2021) (using ETTh2 as a representative), Weather (Wetterstation) and Exchange (Lai et al., 2018). We follow the experimental settings of (Wu et al., 2022b; 2023) where the past sequence length is set to 96, and the prediction length is one of $\{96, 192, 336, 720\}$. We compare the empirical performance with the latest and state-of-the-art Transformer-models including Nonstationary Transformer (Liu et al., 2022), Fedformer (Zhou et al., 2022) and Autoformer (Wu et al., 2022b).

Table 6: Long-term forecasting task on ETTh2, Weather and Exchange.

| Datasets | Prediction Length | Nonstationary (Liu et al., 2022) | | Nonstationary+CAB (Ours) | | Fedformer (Zhou et al., 2022) | | Autoformer (Wu et al., 2022b) | |
|---|---|---|---|---|---|---|---|---|---|
| | | MSE | MAE | MSE | MAE | MSE | MAE | MSE | MAE |
| ETTh2 | 96 | 0.476 | 0.458 | 0.376 | 0.407 | 0.358 | 0.397 | 0.346 | 0.388 |
| ETTh2 | 192 | 0.512 | 0.493 | 0.513 | 0.476 | 0.429 | 0.439 | 0.456 | 0.452 |
| ETTh2 | 336 | 0.552 | 0.551 | 0.522 | 0.486 | 0.496 | 0.487 | 0.482 | 0.486 |
| ETTh2 | 720 | 0.562 | 0.560 | 0.549 | 0.508 | 0.463 | 0.474 | 0.515 | 0.511 |
| **Average** | | 0.526 | 0.516 | 0.490 | 0.469 | 0.437 | 0.449 | 0.450 | 0.459 |
| Weather | 96 | 0.173 | 0.223 | 0.189 | 0.241 | 0.217 | 0.296 | 0.266 | 0.336 |
| Weather | 192 | 0.245 | 0.285 | 0.242 | 0.285 | 0.276 | 0.336 | 0.307 | 0.367 |
| Weather | 336 | 0.321 | 0.338 | 0.307 | 0.333 | 0.339 | 0.380 | 0.359 | 0.395 |
| Weather | 720 | 0.414 | 0.410 | 0.379 | 0.382 | 0.403 | 0.428 | 0.419 | 0.428 |
| **Average** | | 0.288 | 0.314 | 0.279 | 0.310 | 0.309 | 0.360 | 0.338 | 0.382 |
| Exchange | 96 | 0.111 | 0.237 | 0.123 | 0.249 | 0.148 | 0.278 | 0.197 | 0.323 |
| Exchange | 192 | 0.219 | 0.335 | 0.224 | 0.340 | 0.271 | 0.380 | 0.300 | 0.369 |
| Exchange | 336 | 0.421 | 0.476 | 0.327 | 0.416 | 0.460 | 0.500 | 0.509 | 0.524 |
| Exchange | 720 | 1.092 | 0.769 | 0.983 | 0.757 | 1.195 | 0.841 | 1.447 | 0.941 |
| **Average** | | 0.461 | 0.454 | 0.414 | 0.440 | 0.519 | 0.500 | 0.613 | 0.539 |

**Results.** As shown in Table 6, CAB, when integrated with Nonstationary Transformer, consistently improves the performance of the base model on all of the considered datasets spanning different disciplines. Moreover, Nonstationary+CAB even achieves the best performance among the baselines on the Weather and Exchange datasets. This demonstrates the potential of CAB and the mixture-of-head attention design even in MTS predictive tasks.

## D  RUN-TIME ANALYSIS

We further provide performance measurement of the baselines for the imputation task on ETTh (Zhou et al., 2021) (using ETTh1 as a representative). In Table 7, we report the average run-time per iteration (s / iter) of all the methods tested in Section 4.1.

Table 7: Run-time per iteration in (s / iter) for imputation task on ETTh1.

| Series Length | TimesNet (Wu et al., 2023) | Nonstationary (Liu et al., 2022) | Nonstationary+CAB (Ours) | Transformer (Vaswani et al., 2017) | Transformer+CAB (Ours) | FEDformer (Zhou et al., 2022) | DLinear (Zeng et al., 2022) | Autoformer (Wu et al., 2022b) |
|---|---|---|---|---|---|---|---|---|
| 384 | 0.024 | 0.046 | 0.069 | 0.024 | 0.067 | 0.807 | 0.006 | 0.070 |
| 768 | 0.040 | 0.118 | 0.121 | 0.082 | 0.103 | 1.055 | 0.006 | 0.071 |
| 1536 | 0.045 | 0.467 | 0.542 | 0.104 | 0.278 | 1.482 | 0.007 | 0.129 |

**Results.** As shown in Table 7, the CAB incurs only minimal overhead to the base Transformers, especially for the Nonstationary+CAB baseline which achieves the state-of-the-art results for imputation. We note that the two baselines TimesNet and DLinear with superior run-time performance are non-Transformer models and sub-optimal in their achievable error performance.