# OpenReview forum: "Correlated Attention in Transformers for Multivariate Time Series"
_ICLR.cc/2024/Conference — Submitted to ICLR 2024_

### Official Review · Reviewer_rWhs · 2023-10-22

**Soundness:** 3 good
**Presentation:** 3 good
**Contribution:** 3 good
**Rating:** 6
**Confidence:** 2

**Summary:**

The paper introduces a novel "correlated attention mechanism" specifically designed to address the challenges presented by cross-correlations in Multivariate Time Series (MTS). Recognizing a gap in existing Transformer-based models which do not adequately capture these cross-correlations, this research seeks to bridge this gap by offering an advanced mechanism that not only grasps instantaneous cross-correlations but also encompasses lagged cross-correlations and auto-correlation.

The correlated attention mechanism is adeptly crafted to compute cross-covariance matrices between different lag values for queries and keys. A significant feature of this mechanism is its ability to be seamlessly integrated into popular Transformer models, thereby enhancing their efficiency.

In practical applications, such as production planning, the mechanism demonstrates its utility by effectively addressing the lagged interval between variations like demand and production rates. The research further strengthens its case by adapting the original multi-head attention to accommodate both temporal attentions from existing models and the newly proposed correlated attentions. This design ensures that the base Transformer's embedded layer is directly enhanced with cross-correlation information during representation learning.

**Strengths:**

**Strengths**

The authors meticulously focus on harnessing transformer-based architectures for addressing the forecasting problems associated with multivariate time series (MTS). After a thorough investigation of the prevailing methods in the industry, they present a pivotal question:

*How can we seamlessly elevate the broad class of existing and future Transformer-based architectures to also capture feature-wise dependencies? Can modelling feature-wise dependencies improve Transformers’ performance on non-predictive tasks?*

To address this, the authors:

1. Delve deep into the mechanisms of Self-attention and De-stationary Attention. They argue that the Transformer models, as currently conceived, cannot explicitly utilize information at the feature dimension. While there have been efforts to tackle these concerns, the extant methodologies are either too specialized or do not adequately account for the intricacies inherent to MTS data.

2. Introduce the Correlated Attention Block (CAB) as a remedy to the aforementioned challenges. They employ normalization to stabilize the time series and leverage lagged cross-correlation filtering to manage lag-related issues. Furthermore, score aggregation is utilized to consolidate scores from different lagged time points, culminating in the final output.

3. Propose rapid computation techniques for CAB, alongside strategies for its integration into multi-head attention mechanisms.

4. The paper excels in its mathematical exposition – the formulas are presented in a standardized manner, making them easy to follow. Additionally, the experiments are comprehensive and well-executed.

In terms of originality, quality, clarity, and significance, this work shines by offering both a novel perspective and tangible solutions to the MTS forecasting problems using transformer-based architectures. Combining existing ideas with innovative approaches, the paper removes limitations observed in previous results, making it a notable contribution to the domain.

**Weaknesses:**

While the paper has several strengths, there are also areas where it could be improved:

1. Lack of evaluation on prediction tasks: For prediction tasks, such as the MLTSF dataset, the paper does not provide an evaluation of the impact of the Correlated Attention Block (CAB) or compare it with other models that utilize inter-variable correlations. Including such evaluations and comparisons would provide a more comprehensive understanding of the effectiveness of CAB in prediction tasks.

2. Insufficient description of hyperparameter settings: The paper lacks detailed explanations of the hyperparameter settings. For example, in Equation (5), how the initial value of lambda (\lambda) is chosen and how the values of k and c are determined are not clearly stated. Providing more guidance on these hyperparameters would help readers understand the choices made and improve reproducibility.

3. Non-compliance with ICLR submission requirements: The paper does not follow the submission requirements of ICLR by placing the appendix together with the main text. It would be better to separate the appendix from the main text, following the formatting guidelines specified by the conference.

Addressing these areas of improvement would enhance the clarity, reproducibility, and comprehensiveness of the paper, providing readers with a better understanding of the proposed method and its performance in prediction tasks.

**Questions:**

1. How were the hyperparameters determined in the experiments? Specifically, can you provide more details on the selection of hyperparameters such as the initial value of lambda (\lambda) and the values of k and c? Understanding the rationale behind these choices would help in reproducing the results and provide insights into the sensitivity of the proposed method to hyperparameter settings.

2. Can you provide a more detailed evaluation of the Correlated Attention Block (CAB) on prediction tasks, such as the MLTSF dataset? It would be interesting to see how CAB performs compared to other models that utilize inter-variable correlations in prediction tasks. This analysis would shed light on the effectiveness of CAB in different scenarios and provide a better understanding of its potential advantages.

3. In Table 2 and Table 3, it is observed that in some cases, the performance of CAB+Transformer is not as good as Nonstationary, and in some cases, Nonstationary+CAB even leads to worse results. Can you provide an explanation for these observations? What factors contribute to the varying performance of the proposed method in different settings? Understanding the limitations and potential trade-offs of the proposed method would provide valuable insights for future improvements.

---

> ### Author Response · Authors · 2023-11-16
> **Reply 1.1 to Reviewer rWhs**
>
> We are really appreciative of your acknowledgement of our work's contribution. Per your request, we have also included additional experiments in Appendix C in the revised manuscript on long-term forecasting for three common datasets spanning various domains. The results demonstrate the benefit of our CAB even in forecasting. Given this and if the following concerns of yours are addressed, we hope you may consider increasing the score for our work.
>
>  $\textbf{Q1:}$ "Can you provide a more detailed evaluation of the Correlated Attention Block (CAB) on prediction tasks, such as the MLTSF dataset? It would be interesting to see how CAB performs compared to other models that utilize inter-variable correlations in prediction tasks. This analysis would shed light on the effectiveness of CAB in different scenarios and provide a better understanding of its potential advantages."
>
>
>
> $\textbf{Response:}$ As mentioned above, we have included additional experiments for Multivariate Long-term Time Series Forecasting in Appendix C in the revised manuscript. As shown therein, when paired with a strong baseline for long-term forecasting, CAB results in  competitive performance. Specifically, in the dataset like Exchange in our test, CAB gives significant performance boost for longer series sequence. With the flexibility of CAB, we look forward to its deployment on top of many existing Transformers for efficiency improvement.
>
>
>
> $\textbf{Q2:}$ "In Table 2 and Table 3, it is observed that in some cases, the performance of CAB+Transformer is not as good as Nonstationary, and in some cases, Nonstationary+CAB even leads to worse results. Can you provide an explanation for these observations? What factors contribute to the varying performance of the proposed method in different settings? Understanding the limitations and potential trade-offs of the proposed method would provide valuable insights for future improvements."
>
>
> $\textbf{Response:}$ For the anomaly detection task in Table 3, we note that the main metrics within interest is usually the F1 score, which is the harmonic mean of precision (P) and recall (R), thereby giving balance between the two. In terms of F1, CAB actually provides consistent improvement when plugged into both Transformer and Non-stationary Transformer, where decrease in F1 score is only in one dataset for each base model. For the imputation task in Table 2, this task differs from other tasks in that it involves missing data. This can make the representation learning in CAB harder, as the lagged cross-correlation filtering step is sensitive to time step data. For example, if a time lag with high correlation is yet missing, CAB would not have data to detect such potentiall lagged cross-correlation.
>
>
> $\textbf{Q3:}$ How were the hyperparameters determined in the experiments? Specifically, can you provide more details on the selection of hyperparameters such as the initial value of lambda (\lambda) and the values of k and c? Understanding the rationale behind these choices would help in reproducing the results and provide insights into the sensitivity of the proposed method to hyperparameter settings.
>
>
>
> $\textbf{Response:}$ We kindly refer the reviewer to our Appendix A in the paper for details on implementation and hyperparameter setting, which would provide information to all of your question.
>
>
> $\textbf{Q4:}$ "Non-compliance with ICLR submission requirements: The paper does not follow the submission requirements of ICLR by placing the appendix together with the main text. It would be better to separate the appendix from the main text, following the formatting guidelines specified by the conference."
>
>
> $\textbf{Response:}$ We thank you for that, and will separate the main text from the supplementaries in later versions including our revised manuscript now.
>
> ------
>
> We thank you again for your meaningful comments. If there is any concern left, we would be pleased to answer in the remaining time of the discussion period. If our positive results on forecasting as well as asnwers could have properly addressed your main concerns, we hope that you could consider raising the score for our work.

---

> > ### Author Response · Authors · 2023-11-19
> > **Discussion Period**
> >
> > If you have further concerns regarding our answers, we are more than happy to respond to them during this discussion period. Otherwise, we would really appreciate your acknowledgement that you concerns indeed have been cleared.
> >
> > Again, we thank you very much for your time and reviews!

---

> > > ### Author Response · Authors · 2023-11-22
> > > **End of Discussion Period**
> > >
> > > Dear Reviewer,
> > >
> > > We kindly remind that the discussion period is coming to the end. Thus, please provide us with any concern you may still have. Otherwise, we would really appreciate your acknowledgement that you concerns indeed have been cleared and hope that you consider raising the score for our work, especially given the additional experiments having been added supporting our method.
> > >
> > > Again, we thank you very much for your time and and hard work reviewing our paper!
> > >
> > > Authors

---

> > > > ### Author Response · Authors · 2023-11-22
> > > > **Reminder till the end of discussion period**
> > > >
> > > > Dear Reviewer,
> > > >
> > > > This is a gentle reminder for the case of our paper, as it is  less than 24 hours till the end of discussion period.
> > > >
> > > > Please kindly note that Reviewer mjBo has very positively increased the score to acceptance threshold, after we provided new long-term forecasting results on the high-dimensional Traffic dataset; more details can be found in our response therein to him/her.
> > > >
> > > > Up till now, we have provided comprehensive results on 4 datasets for long-term forecasting showing the benefit and state-of-the-art Transformer results from CAB on 3 out of 4 tested datasets even on this task, and empirically shown that our CAB module incurs very minimal computational overhead (for both high-dimensional data and scaling input series length).
> > > >
> > > > Furthermore, we will publish the code, if the paper may get accepted. Thanks to the modularity of CAB, which can be seamlessly integrated into existing base Transformers, we believe that this will be beneficial to the community.
> > > >
> > > > In view of the very good additional experiments aligning with your main question and if your concerns have been all properly clarified, we really hope that you can positively increase the score for our work.
> > > >
> > > > Thank you again for your time and review of the paper, which has helped us to improve our work.
> > > >
> > > > Authors

---

> > > > > ### Author Response · Authors · 2023-11-23
> > > > > **Final reminder and thank you note to Reviewer rWhs**
> > > > >
> > > > > Dear Reviewer rWhs,
> > > > >
> > > > >
> > > > > As there are only a few hours left, we would like to take this chance to note that two reviewers have positively increased the scores to acknowledge our clarification of the general concerns.
> > > > >
> > > > > We are much appreciative of your recognition of our work's contributions and constructive comments, especially on the request for the CAB evaluation on long-term forecasting. To this end, our additional experiments on long-term forecasting on 3 datasets (with additional 1 in our reply to mjBo) have been integrated into our manuscript to improve the paper, and indeed further confirm the benefits of our CAB.
> > > > >
> > > > > We have also referred you to our Appendix A in the manuscript for our hyperparameter settings and will further release the code if the paper may get accepted. We believe that the modularity and robustness of CAB will be beneficial to improving a wide class of Transformers, thereby being helpful for the community.
> > > > >
> > > > > Given all these, we really hope that you may consider increasing the score of our paper. We thank you very much again for your time and hard work to review this paper.
> > > > >
> > > > > Best regards,
> > > > >
> > > > > Authors

---

### Official Review · Reviewer_KQRi · 2023-10-31

**Soundness:** 2 fair
**Presentation:** 3 good
**Contribution:** 2 fair
**Rating:** 5
**Confidence:** 3

**Summary:**

The paper focuses on how to learn the feature-wise correlation when applying a transformer in the multivariate time series for various tasks.  The proposed correlated attention operates across feature dimensions to compute a cross-variance matrix between keys and queries. They introduce a lag value in the process so that it can learn not only instantaneous but also aged cross-correlations. The proposed method shows improved performance on tasks such as classification and anomoly detection.

**Strengths:**

The paper's main focus is to address the learning of feature-wise correlation in the transformer attention setup. They explore if the learning feature-wise correlation actually helps in tasks other than forecasting such as anomaly detection, imputation, and classification.

The proposed correlated attention can capture not only conventional cross-correlation but also capture auto-correlation, and lagged cross-correlation.  The idea that allows one to learn lagged correlation and be able to integrate the most relevant multiple lagged correlation sounds interesting.

**Weaknesses:**

1, Some parts of the paper presentation could be improved,  such as the explanation of the methods, for more details check the question sections.
2. The experiment section does not look very convincing due to the comparison setup (if it is fair or not, please refer to the question section) and results. Given the huge computational cost of integrating the cross-correlation, the experiment results do not look that significant.

**Questions:**

1. Some parts are a bit confusing, for instance,
“CrossFormer deploys a convoluted architecture, which is isolated from other prevalent Transformers with their own established merits in temporal modeling and specifically designed for only MTS forecasting, thereby lacking flexibility” I am a bit confused, could you explain more in detail this?

The section to explain equation 5 needs to be improved. Especially the explanation for operator argTopK()  reads a bit confusing.

3. It is confusing how the value of k in equation 6 is defined, do you get the value of c first and then calculate k with the topK operation?  It is not very clear to me why not directly take top k,  for instance, top 5,  lagging value, and use it instead of getting a value k by using the topK operator?  Any motivation behind?

4. I am not sure how to go from equation 7 to the result they got in the section below, maye some proof?
5. It seems even with FFT, the computational complexity is still quite high for a time series with a large feature dimension.

6. When evaluating a specific task, it is crucial to compare its performance with a model that has been explicitly designed and optimized for that particular task. For instance, both non-stationary transformers, dlinear and FEDformer are designed for forecasting tasks. The reviewer are not 100% sure that whether it is a fair comparison when applying those to classification, and anomaly detection tasks.

7. I think the most fair comparison is transformer vs transformer +CBA where the transformer has the same number of heads as the transformer +CBA (when we count both temporal attention and correlated attention heads). Does the transformer in Table 2 has exact same head as the transformer +CBA? The results of the transformer in that table do not show much improvement when compared to transformer + CBA.

8. In anomaly detection and classification, it shows that transformer +CBA has significant improvement compared to transformer, this was not observed in the imputation task, any insights into that?

9. It would be interesting to see the performance on the forecasting task as well I think since there is nothing in the design that specifically restricts it to the non-predictive task.

10 . I think the paper main motivation was addressing the feature-wise correlation specifically for non-predictive tasks, but I am missing the discussion what is the difference when you learning the feature-wise correlation for forecasting or for non-predictive task and what is in this model that makes it more fit for the non-predictive task

---

> ### Author Response · Authors · 2023-11-16
> **Reply 1.1 to Reviewer KQRi**
>
> We thank you the reviewer for all the comments and suggestions, and would like to hereby address your questions. If we could answer all of your concerns, we hop taht you can consider increasing the score for our work.
>
>
> $\textbf{Q1:}$ " “CrossFormer deploys a convoluted architecture, which is isolated from other prevalent Transformers with their own established merits in temporal modeling and specifically designed for only MTS forecasting, thereby lacking flexibility” I am a bit confused, could you explain more in detail this?"
>
> $\textbf{Response:}$ We just want to discuss that CrossFormer's architecture is very unconventional from other prevalent Transformer with simple multi-head attention that yet has been  shown to be very good in temporal modeling. By deviating from the very base architecture of vanilla Transformer, Crossformer may not utilize such advantages; for your information, Crossformer deploys a Hierarchical Encoder-Decoder architecture with Dimension-Segment-Wise embedding, which devides time into segmants,  Two-Stage Attention and  segment merging, which merges back the time. We thank you for the suggestion, and will re-write it for more clarity in the updated versions in the future.
>
>
>
> $\textbf{Q2:}$ "It is confusing how the value of k in equation 6 is defined, do you get the value of c first and then calculate k with the topK operation? It is not very clear to me why not directly take top k, for instance, top 5, lagging value, and use it instead of getting a value k by using the topK operator? Any motivation behind?"
>
> $\textbf{Response:}$ We would like to clarify that such $k = c \log(T)$ is indeed the input $k$ to TopK operation, instead of getting a value $k$ out of topK operator. So yes, you would choose $c$ first, which is a hyperparameter, and then use $k = c \log(T)$ as the input to TopK. Also, motivation for doing $k = c \log(T)$ is that even if one uses FFT  of $O(T \log(T))$ for fast computation, your final computational cost will still depend on $O(d^2 T \cdot k  )$ for summing up $k$ terms in Eq. (6), so setting $k = c \log(T) = O(\log(T))$ would allow you to maintain such $O(T \log(T))$ complexity.
>
>
> $\textbf{Q3:}$ "I am not sure how to go from equation 7 to the result they got in the section below, maye some proof?"
>
> $\textbf{Response:}$ We would like to explain it in more details here. First, please note that Eq (7)
> $S_{X, Y}(f) = F(X_t) F^*(Y_t)$ should be interpreted as $S_{X, Y}(f) = F(X_t)(f) \cdot F^*(Y_t)(f)$. Note that in the two equations in Eq Eq (7), the variable $f$ corresponds to frequency domain and $l$ corresponds to time domain. While we write $S_{X, Y}(f)$  and $(X \star Y)(l)$ in point-wise form as in the paper, which demonstrates the Cross-correlation Theorem and how you can compute  $(X \star Y)(l)=  \sum_{t=1}^{T} X_{t-l} Y_t$ via FFT and inverse FFT, the actual implementation using Pytorch's FFT library would let you input, for example in case of $d=1$ feature, the vector $X_t$  of size $T$ in time domain (i.e. a vector of all $l$ stacked together), and perform FFT in $O(T \log(T))$ to return you the vector $F(X_t)$ of size $T/2+1$ in frequency domain (i.e. a vector of all $f$ stacked together). From there, you can compute $S_{X, Y}(f) = F(X_t) F^*(Y_t)$, which is again a vector  of all $f$ stacked together.  From the second equation in Eq (7), you can now  feed such vector into the inverse FFT to get the vector  $(X \star Y)(l)= F^{-1}( S_{X, Y}(f) )$ of all $l = 0\to T-1$ stacked together. We hope that this can clarify your concern.
>
>
> $\textbf{Q4:}$ "It seems even with FFT, the computational complexity is still quite high for a time series with a large feature dimension."
>
> $\textbf{Response:}$ We kindly refer the reviewer to our new experiments on run-time of the baselines in Appendix D of the manuscript. It shows that the overhead incurred by our CAB to the base models is quite minimal, so the resulting models still maintain around the same order of magnitude in computation as other SOTA Transformers. We do agree with you CAB will have another multiplicative factor of $d$ in its complexity, yet CAB has advantage in $T$, which has been usually the real computational bottleneck to be addressed in the literature of Transformers' computational complexities. Further acceleration in $d$ would be interesting future direction.

---

> > ### Author Response · Authors · 2023-11-16
> > **Reply 1.2 to Reviewer KQRi**
> >
> > $\textbf{Q5:}$ "When evaluating a specific task, it is crucial to compare its performance with a model that has been explicitly designed and optimized for that particular task. For instance, both non-stationary transformers, dlinear and FEDformer are designed for forecasting tasks. The reviewer are not 100% sure that whether it is a fair comparison when applying those to classification, and anomaly detection tasks."
> >
> > $\textbf{Response:}$ We kindly note that, as mentioned in our original submission, Nonstationary Transformer and FEDformer are in fact the state-of-the-art (SOTA) Transformer-models for respectively imputation and anomaly detection in the recent benchmarks (Wu et al., 2023).
> > Yes, while they were designed for forecasting, their architectures were quite robust and good enought to achieve competitive results on even encoder-only tasks. For the overall SOTA, it has been TimesNet for all of these tasks.
> > In our revised manuscript, for classficiation task, we have further added Flowformer, which is the SOTA Transformer for classification on these 10 datasets, for completeness. So our experiments for all the 3 encoder-only tasks now include an overall SOTA baseline and a SOTA Transformer for each of the task.
> >
> >
> >
> > $\textbf{Q6:}$ "I think the most fair comparison is transformer vs transformer +CBA where the transformer has the same number of heads as the transformer +CBA (when we count both temporal attention and correlated attention heads). "
> >
> > $\textbf{Response:}$ As discussed in our Appendix A of the original submission, we use 8 heads for Transformer or Nonstationary Transformer, and 16 heads for our combined models. We keep 8 heads only because it follows the benchmark that most if not all related work have done. At the same time, increasing the number of heads would not necessarily improve the overall accuracies as it also comes with the burden of more parameters to be learnt. For classification, we further provide additional experiments below for Transformer and Nonstationary Transformer with 16 heads. As you can see, the variation is very minimal between 8 heads and 16 heads, and does not affect the relative rankings between the baselines.
> >
> >
> >
> > | Datasets      | Transformer (8 heads) | Transformer (16 heads) | Nonstationary (8 heads) | Nonstationary (16 heads)|
> > | ----------- | ----------- |----------- | ----------- | ----------- |
> > | Ethanol      |     26.24   |  31.94   | 25.10   | 25.10
> > | FaceDetection | 67.93       |     69.06 |   69.40| 68.70
> > |    Handwriting | 29.53        |   28.23   | 30.12      |  26.47
> > |  Heartbeat| 75.12  |     71.70    | 72.20 |      72.19
> > |   JapaneseVowels | 97.03 |  97.03       | 95.68     | 96.75
> > |   PEMS-SF | 78.03 |      81.50   |    75.14   |     75.14
> > |   SCP1 | 91.13 |     89.07    |     82.94 |  85.67
> > |   SCP2 | 53.89 |      55.55   |   55.55  |     55.00
> > |SpokenArabic | 98.45    |    98.63     |    98.91   |    98.59
> > |    UWaveGesture | 86.25|       82.50  |    85.94  |     85.00
> > |   Average |70.36 |     70.52    |     69.10   | 68.86
> >
> > $\textbf{Q7:}$ "In anomaly detection and classification, it shows that transformer +CBA has significant improvement compared to transformer, this was not observed in the imputation task, any insights into that?"
> >
> > $\textbf{Response:}$ We believe that this may come from two reasons. First, unlike the other tasks, imputation involves missing data, which makes the learning of \emph{lagged} cross-correlation, which is sensitive to time steps, harder. Second, the Transformer baseline itself is not good or compatible for imputation, where you can see on some datasets like ETTm2, ETTh1, ETTh2 and Electricity, its error is order of magnitude larger than the SOTA baselines.

---

> > > ### Author Response · Authors · 2023-11-16
> > > **Reply 1.3 to Reviewer KQRi**
> > >
> > > $\textbf{Q8:}$ "It would be interesting to see the performance on the forecasting task as well I think since there is nothing in the design that specifically restricts it to the non-predictive task.";
> > > "I am missing the discussion what is the difference when you learning the feature-wise correlation for forecasting or for non-predictive task and what is in this model that makes it more fit for the non-predictive task"
> > >
> > >
> > > $\textbf{Response:}$ Please find our new experiments on long-term forecasting on three datasets spanning different domains in Appendix C of the paper. CAB demonstrates consistent performance boost when plugged into Non-stationary Transformer as the base model. It in fact outperforms  recent SOTA Transformer models designed for long-term forecasting.
> > >
> > > For forecasting, we believe one challenge to fully utilize the potential of CAB is to have a proper design of the masking mechanism. First, we recall the main difference between the decoder (used in forecasting) and the encoder is that the decoder has two multi-head attention (MHA) blocks: one with masking and one without masking (i.e. conventional MHA). The masking is meant to prevent information to preserve the auto-regressive property, so that the forecasting of the next time step should depend only on the time steps strictly before it. In typical self-attention mechanism, this is done by  masking out (setting to $-\infty$) all values in the lower triangular of the scoring matrix $Q K^T\in \mathbb{R}^{T \times T}$ that is input to softmax. As you can see, $Q K^T$ can be thought of as the attention/correlation matrix in time (with dimension $T\times T$), so masking out the triangular part corresponds to "blinding" the history in making predictions; one the other hand, in CAB, the $Roll(K, l)^T Q \in \mathbb{R}^{d_k \times d_k}$, which is input to softmax, is correlation matrix across feature with dimension $d_k \times d_k$ instead of time, so the same masking strategy, even if naively applicable, would not have intuitive or physical phenomena as in vanilla self-attention, thereby being more of a heuristic approach. In the experiments in Appendix C, we only replace the non-masked MHA with our mixture-of-head attetions comprised of temporal attentions and our CAB, while keeping the masked MHA of the base Transformer. As you can see from the results, the CAB consistently improves performance of Non-stationary Transformer across all the tested datasets and resulting in the model that outperforms the previous SOTA Transformers for long-term forecasting.
> > >
> > >
> > > ------
> > >
> > > We much appreciate your detailed review with insightful comments. If there is any remaining question, we will be pleased to address it in the remaining time of the rebuttal phase.

---

> > > > ### Author Response · Authors · 2023-11-19
> > > > **Discussion Period**
> > > >
> > > > If you have further concerns regarding our answers, we are more than happy to respond to them during this discussion period. Otherwise, we would really appreciate your acknowledgement that you concerns indeed have been cleared.
> > > >
> > > > Again, we thank you very much for your time and reviews!

---

> > > > > ### Author Response · Authors · 2023-11-22
> > > > > **End of Discussion Period**
> > > > >
> > > > > Dear Reviewer,
> > > > >
> > > > > We kindly remind that the discussion period is coming to the end. Thus, please provide us with any concern you may still have. Otherwise, we would really appreciate your acknowledgement that you concerns indeed have been cleared and hope that you consider raising the score for our work, especially given the additional experiments having been added supporting our method.
> > > > >
> > > > > Again, we thank you very much for your time and and hard work reviewing our paper!
> > > > >
> > > > > Authors

---

> > > > > > ### Author Response · Authors · 2023-11-22
> > > > > > **Reminder till the end of discussion period**
> > > > > >
> > > > > > Dear Reviewer,
> > > > > >
> > > > > > This is a gentle reminder for the case of our paper, as it is  less than 24 hours till the end of discussion period.
> > > > > >
> > > > > > Please kindly note that Reviewer mjBo has very positively increased the score to acceptance threshold, after we provided new long-term forecasting results on the high-dimensional Traffic dataset; more details can be found in our response therein to him/her.
> > > > > >
> > > > > > Up till now, we have provided comprehensive results on 4 datasets for long-term forecasting showing the benefit and state-of-the-art Transformer results from CAB on 3 out of 4 tested datasets even on this task, and empirically shown that our CAB module incurs very minimal computational overhead (for both high-dimensional data and scaling input series length).
> > > > > >
> > > > > > Furthermore, we will publish the code, if the paper may get accepted. Thanks to the modularity of CAB, which can be seamlessly integrated into existing base Transformers, we believe that this will be beneficial to the community.
> > > > > >
> > > > > > If your concerns have been all properly clarified, we hope that you can positively re-evaluate our work up to the acceptance threshold.
> > > > > >
> > > > > > Thank you again for your time and review of the paper, which has helped us to improve our work.
> > > > > >
> > > > > > Authors

---

> > > > > > > ### Author Response · Authors · 2023-11-23
> > > > > > > **Final reminder and thank you note to Reviewer KQRi**
> > > > > > >
> > > > > > > Dear Reviewer KQRi,
> > > > > > >
> > > > > > > As there are only a few hours left, we kindly note that two reviewers have positively increased the scores for our work to 6 to acknowledge our clarification of the concerns, many of which align with your main questions about CAB's computational overhead and performance on prediction tasks.
> > > > > > >
> > > > > > > We really hope that you can positively re-evaluate our work, if all of your concerns are addressed and in view of our additional experiments, all of which further demonstrate benefits of CAB and have been integrated directly into our paper. If you may have any question left, we will do our best to response in these last hours.
> > > > > > >
> > > > > > > Again, we thank you very much for your time and hard work spent on reviewing this paper. Besides the experiments on run-time of baselines and forecasting that are now in the revised manuscript, we will also include experiments on Transformer and Non-stationary Transformer with 16 heads in our reply to you in our future manuscript for better clarity and more comprehensive comparision.
> > > > > > >
> > > > > > > We are appreciative of all your constructive comments and hope that you can give acknowledgement to our effort addressing your concerns during the rebuttal.
> > > > > > >
> > > > > > > Thank you very much and best regards,
> > > > > > >
> > > > > > > Authors

---

> > > > > ### Comment · Reviewer_KQRi · 2023-11-23
> > > > > **Reviewer comment**
> > > > >
> > > > > Thank you for the author's detailed answer, clarification, and extra experimental results. After going through the revision, and the other reviewers' comments and answers, I decided to keep my score. My reasons are that compared to the computational complexities, the introduced method seems to have very minor improvement, I am skeptical of how much of that contribution is really coming due to the introduction of the feature-wise correlation or if it is due to other factors such as optimizations, randomness and some differences on the architects of the network.

---

### Official Review · Reviewer_mjBo · 2023-10-31

**Soundness:** 2 fair
**Presentation:** 3 good
**Contribution:** 1 poor
**Rating:** 6
**Confidence:** 3

**Summary:**

The author extend autoformer to cross-correlation and propose a correlated attention mechanism to capture feature-wise dependencies.

**Strengths:**

1. The authors propose a correlated attention mechanism to capture lagged cross-covariance between variates, which can combined with existing encoder-only transformer structure.
2. The experiments show correlated attention mechanism enhances base Transformer models.

**Weaknesses:**

1. The novelty of the paper is limited.  The proposed correlated attention is basically a extension for Autoformer, which only captures auto-correlation. However, this method neither proposes a good method to reduce the computational complexity caused by calculating cross-correlation, which is almost unacceptable in actual scenarios, nor does the author conduct a comparative experiment with Autoformer to prove that the introduction of corss-correlation can bring to achieve practical improvements.
2. The integration of correlated attention to existing transformer structure is conducted with a mixture-of-head attention structure, which is a concatenation of CAB and transformer outputs. CAB acts as a rather independent component and does not truly integrated into existing transformer structures.

**Questions:**

1. Judging from the design of CAB, it can predict independently without requiring additional transformer deconstruction. Why didn't you test the independent CAB?
2. The design of CAB is based on autoformer. Why is there no comparison between the effects of CAB and autoformer?

---

> ### Author Response · Authors · 2023-11-16
> **Reply 1.1 to Reviewer mjBo**
>
> We thank the reviewer for the comments, and believe that there can be some misunderstanding on our work contribution that we would be pleased to hereby clarify.
> We really hope that if your concerns can be properly addressed in the rebuttal and the position of our contribution is clarified, you can positively re-evaluate our work. Before giving our detailed answers, we kindly summarize the following points:
> - Our architecture has many differences from Autoformer, which itself was very specifically designed to capture auto-correlation in long-term forecasting, in order to capture lagged cross-correlation. To this end, we have provided a paragraph (highlighted in blue) at the end of Section 3.2.2 in the revised manuscript to emphasize the differences.
>
> - We have included the performance of Autoformer in all of our experiments in the revised manuscript. The results confirm that Autoformer is not as competitive as other baselines and our models; this is quite expected as Autoformer was designed for forecasting, and also a reason why we did not include it in our original experiments focusing more on encoder-only tasks.
> - For completeness, in Appendix C of the revised manuscript, we also provide new experiments on long-term forecasting, which Autoformer excels in, on three datasets spanning different domains. In these tests, the our Non-stationary+CAB model achieves the best performance among Transformer on two out of three datasets. While Autoformer still has very competitive results, it generally lags behind Fedformer, which is one of the latest and SOTA Transformers for long-term forecasting.
>
> $\textbf{Q1:}$ "The novelty of the paper is limited. The proposed correlated attention is basically a extension for Autoformer, which only captures auto-correlation. ";
> "The design of CAB is based on autoformer. Why is there no comparison between the effects of CAB and autoformer?"
>
>
> $\textbf{Response:}$ We believe that our proposed CAB (correlated attention block) has many differences from the Autoformer, and have provided a paragraph (highlighted in blue) at the end of Section 3.2.2 in the revised manuscript to emphasize the differences. We would like to next re-iterate our discussion in the paper here in our response for your convenience.
>
> Since the CAB aims to capture the lagged cross-correlation,which is relevant to yet more generalized than the auto-correlation module in Autoformer, we believe it is crucial to emphasize the main differences. First, Autoformer overall is a decomposed encoder-decoder architecture proposed for long-term forecasting, so its auto-correlation module is specifically designed to work with series seasonalities extracted from various series decomposition steps of Autoformer. On the other hand, CAB ensures flexibility with any input series representation by deploying normalization step and learnable temperature coefficient $\tau$ reweighting the correlation matrices. Second, while Autoformer computes purely auto-correlation scores and aggregates their exact values for TopK , CAB computes cross-correlation matrices and aggregates the absolute values of such entries for TopK in Equation 5 (as correlation can stem from either positive or negative correlation). Finally, to facilitate robustness to different input series representation, CAB adopts learnable weights $\lambda$ in TopK operation, which balances between auto-correlation and cross-correlation, and $\beta$ in sub-series aggregation, which balances between instantaneous and lagged cross-correlation.

---

> > ### Author Response · Authors · 2023-11-16
> > **Reply 1.2 to Reviewer mjBo**
> >
> > $\textbf{Q2:}$ "However, this method neither proposes a good method to reduce the computational complexity caused by calculating cross-correlation, which is almost unacceptable in actual scenarios, nor does the author conduct a comparative experiment with Autoformer to prove that the introduction of corss-correlation can bring to achieve practical improvements."
> >
> > $\textbf{Response:}$ We believe that the CAB indeed very consistently results in practical improvement of the base models, thereby resulting state-of-the-art (SOTA) models on different tasks, and incurs minimal overhead.
> >
> > First, regarding the practical improvements in application performance, we have included Autoformer as a baseline in all of the experiments in our revised manuscript. Then we provided new experiments on long-term forecasting in Appendix C of the paper. In these tests, Autoformer did have competitive performance in imputation task on the Weather dataset (yet still is worse than TimesNet). Other than that, our CAB , when combined with some base Transformer, still demonstrates clear improvement and achieves SOTA results on encoder-only tasks. For forecasting, ours also perform better than other Transformer baselines, including Autoformer, on two out of three datasets under benchmark.
> >
> > Second, regarding computation complexity, we provide new run-time measurements in Appendix D of the revised manuscript to show that the overhead incurred by CAB is in fact very minimal. We also concur with you that CAB indeed has a multiplicative factor of dimension $d_k$ in its complexity compared to other attention mechanisms, yet we have better dependency on series length $T$, which is usually the computational bottleneck and thus the focus of most of the work on Transformers' computational complexity, and observe (via experiments as in Appendix D) that in practice the overhead of CAB is still small thanks to all the required calculations being matrix operations and thus well parallelized in bulit-in library (like Pytorch).
> >
> >
> > $\textbf{Q3:}$ "The integration of correlated attention to existing transformer structure is conducted with a mixture-of-head attention structure, which is a concatenation of CAB and transformer outputs. CAB acts as a rather independent component and does not truly integrated into existing transformer structures."
> >
> >
> > $\textbf{Response:}$ In fact, we believe that this modularity of CAB is its crucial benefit, since it can be seamessly integrated with many existing architectures. In order to facilitate such flexibility and robustness to any input series representation (by the base Transformer models), we leverage many sub-components for representation learning in CAB design: normalization step and learnable parameters at the core steps like $\tau$ for normalizing correlation matrices, $\lambda$ for balancing auto-correlation and cross-correlation, and $\beta$ for balancing instantaneous and lagged cross-correlation.
> >
> >
> > $\textbf{Q4:}$ "Judging from the design of CAB, it can predict independently without requiring additional transformer deconstruction. Why didn't you test the independent CAB?"
> >
> > $\textbf{Response:}$ In fact, we did test the independent CAB in our original submission's Appendix. We kindly refer the reviewer to our ablation studies in Appendix B. It is the pure-CAB-Transformer baseline, which purely uses CAB (i.e. no temporal attention from any base Transformer) and even disables lagged cross-correlation filtering. pure-CAB-Transformer itself also resulted improvement over Autoformer.
> >
> > ------
> > We really believe that there is some misunderstanding and unfortunate inclarity in our work's contributions. We hope that our answers, supported by the provided new experiments adding Autoformer and considering long-term forecasting (Autoformer's most advantageous task), would help clarify your concerns. If your concerns are cleared, please kindly positively re-evaluate our work. We would be pleased to address any remaining concern.

---

> > > ### Author Response · Authors · 2023-11-19
> > > **Discussion Period**
> > >
> > > If you have further concerns regarding our answers, we are more than happy to respond to them during this discussion period. Otherwise, we would really appreciate your acknowledgement that you concerns indeed have been cleared.
> > >
> > > Again, we thank you very much for your time and reviews!

---

> > > ### Comment · Reviewer_mjBo · 2023-11-20
> > >
> > > Given your experiment on long-term forecasting and the validation of utilizing cross-correlation to achieve further improvements on top of the Autoformer, I would consider boosting my scores. However, I still have some concerns regarding your updated experimental results:
> > >
> > > 1. Currently, there is a limited number of datasets being tested for forecasting tasks, with only three datasets available. Furthermore, in one of these datasets, CAB did not surpass Autoformer. I believe such results make it challenging to demonstrate significant overall improvements brought about by the CAB method with cross-correlation.
> > >
> > > 2. Your runtime analysis was conducted on the ETTh1 dataset, which contains only seven variables. On this dataset, the squared complexity of CAB may not pose a significant issue. However, if applied to datasets with a larger number of variables, such as traffic data, complexity could become a problem.

---

> > > > ### Author Response · Authors · 2023-11-22
> > > > **Reply 2 to Reviewer mjBo**
> > > >
> > > > We thank you for your comments. In order to reply to both of your concerns, we provide the following additional experiments on long-term forecasting on your requested Traffic dataset, which i) asserts that computation is indeed not the bottleneck of our method even on high-dimensional dataset, and ii) provides another example of CAB's benefits in boosting accuracy/performance in long-term forecasting. For illustration, we conduct testing on three baselines Nonstationary Transformer, Nonstattionary Transformer+CAB, and Autoformer.
> > > >
> > > > Specifically, we fix the  prediction length to be 96, and vary the the past sequence length $\in$ { 96, 192, 384} and report the average run-time per iteration (s / iter) in the below table.
> > > >
> > > >
> > > > | Past sequence length    | Nonstationary | Nonstationary+CAB | Autoformer|
> > > > | ----------- | ----------- |----------- | ----------- |
> > > > | 96      |  0.063 | 0.091  |  0.088  |
> > > > | 192      |  0.079   | 0.123 |  0.116  |
> > > > | 384      |   0.126  | 0.238    | 0.170   |
> > > >
> > > > The results suggest that CAB  incurs minimal overhead and will not suffer from computation bottleneck in any dimension (number of features, and scaling series length). The reason is that all of the calculations of CAB are matrix operations and thus can be efficiently implemented ( in tensor forms with high parallelization) in Pytorch. Next, we further report the average MSE and MAE across different past sequence lengths in the next table, therby confirming that CAB consistently boosts performance of base Transformer also on Traffic dataset in long-term forecasting. The resulting performance is also better than Autoformer.
> > > >
> > > > |Average Error    | Nonstationary | Nonstationary+CAB | Autoformer|
> > > > | ----------- | ----------- |----------- | ----------- |
> > > > | Average MSE      |   0.618   |  0.606   |    0.674 |
> > > > | Average MAE      | 0.352    |  0.333  |  0.411  |
> > > >
> > > > Given that our work is clearly stated to focus more on encoder-only tasks (due to the masking mechanism being proper future direction), we believe that the advantages of CAB in long-term forecasting shown in 4 different datasets up till now are comprehensive enough in view of many other tasks/applications tested in the paper.
> > > >
> > > > We hope that your remaining concerns are cleared and will be pleased to do our best addressing any question left in the discussion period. If the contributions of the paper are better clarified now, we hope that the reviewer can positively re-evaluate our work up to the acceptance threshold. Thank you again for your time and review.

---

> > > > > ### Author Response · Authors · 2023-11-22
> > > > > **Thank you note**
> > > > >
> > > > > Dear Reviewer mjBo,
> > > > >
> > > > > We thank you very much for positively increasing the score to 6 for our paper and having been involved with us during the discussion period. Thanks to your comments on the inclusion of Autoformer (both in discussion and experiments), computations and forecasting tasks, our manuscript has been improved and better presented our contribution with more clarity. We much appreciate that.
> > > > >
> > > > > Thank you again and best regards,
> > > > >
> > > > > Authors

---

### Official Review · Reviewer_M5Eq · 2023-11-01

**Soundness:** 3 good
**Presentation:** 3 good
**Contribution:** 2 fair
**Rating:** 6
**Confidence:** 3

**Summary:**

This paper introduces a novel concept called the Correlated Attention Block (CAB), designed to efficiently capture cross-correlations between multivariate time series (MTS) data within Transformer-based models. The CAB is a versatile component that seamlessly integrates into existing models. Its key innovation lies in the correlated attention mechanism, which operates across feature channels, enabling the computation of cross-covariance matrices between queries and keys at various lag values. This selective aggregation of representations at the sub-series level opens the door to automated discovery and representation learning of both instantaneous and lagged cross-correlations while inherently encompassing time series auto-correlation.

The authors conducted an extensive series of experiments, focusing on Imputation, Anomaly Detection and Classification tasks. Their results demonstrate remarkable performance, underscoring the potential of the CAB to enhance the analysis and modeling of MTS data.

**Strengths:**

This paper is well-structured, presenting a thorough background introduction and a step-by-step introduction of the novel concept, the Correlated Attention Block (CAB). A notable feature of this work is the seamless integration of CAB into encoder-only architectures of Transformers, making it a potentially good-to-have addition to the field.

Furthermore, the authors conducted an extensive set of experiments across three different tasks, utilizing a variety of common datasets. The results consistently show impressive performance, often outperforming previous state-of-the-art methods. This robust evaluation underscores the potential of the proposed design in improving representation learning for multivariate time series, making it a valuable contribution to the field.

**Weaknesses:**

1. Page 9, Line 1 of **Conclusion And Future Work**: There's a minor typo that needs correction - "bloc" should be changed to "block."
2. Citation Style: The reference list shows some inconsistency in the citation style. To enhance clarity and uniformity, consider standardizing the format across all references. For example, you could list all NeurIPS papers with consistent formatting, and for papers from other conferences or sources, ensure that their respective publication details are included appropriately. For instance:
    -   NeurIPS papers should consistently include the conference name and URL. For example, "Vaswani et al. (2017, NeurIPS, URL) and Shen et al. (2020, NeurIPS, URL)."
    -   Papers from other conferences or sources should similarly follow a consistent format, such as including the conference name and URL as needed. For example, "Cao et al. (2020, Conference Name, URL)" and "Li et al. (2019, Conference Name, URL)."
3.  Motivation for Correlated Attention Block: While the paper mentions that CAB efficiently learns feature-wise dependencies, it would be beneficial to provide more clarity on the specific aspects of CAB's design that contribute to this efficiency. Clearly articulating which components within the CAB block are the key drivers of this efficiency could help readers better understand the innovation.
4.  Analysis of FFT Efficiency: While Section 3.2.2 discusses the time efficiency of FFT, it would be valuable to include a clear analysis that quantifies how much the use of FFT improves the performance of CAB compared to vanilla CAB or previous baselines. Providing concrete numbers or performance metrics would strengthen the paper's findings in this regard.
5. Limitation of Encoder-Only Models: It's important to acknowledge that the design of CAB is limited to encoder-only models and does not support time series forecasting. While this limitation is briefly mentioned, expanding on the reasons behind this constraint and discussing potential avenues for future work or extensions to address this limitation would add depth to the paper.

**Questions:**

1.  In Table 2, it is evident that for the first three datasets, the TimesNet baseline consistently outperforms CAB when the mask ratio exceeds 25%. Could you provide insights into this performance discrepancy?
2.  Could you elaborate on the primary challenges or obstacles preventing the integration of CAB into encoder-decoder models for conducting multivariate time series forecasting?
3.  Could you provide a detailed breakdown of the distinct contributions of each component within CAB, both in terms of performance enhancement and efficiency gains?

---

> ### Author Response · Authors · 2023-11-16
> **Reply 1.1 to Reviewer M5Eq**
>
> We thank the reviewer very much for the insightful comments as well as the acknowledgement of our proposed method's benefits. Please find our revised manuscript taking into account your suggestions and providing additional experiments in the appendix to address the reviewer's concerns on computational complexity of CAB and CAB's performance on predictive tasks.
> Given the new  positive experimental results and if we can properly address all of your concerns, we really hope that you can increase the score for our work.Now, we hereby would like to address the main concerns of the reviewer.
>
>
>
> $\textbf{Q1:}$ "It would be beneficial to provide more clarity on the specific aspects of CAB's design that contribute to this efficiency.";
> "Could you provide a detailed breakdown of the distinct contributions of each component within CAB, both in terms of performance enhancement and efficiency gains?"
>
> $\textbf{Response:}$ We kindly refer the reviewer to the ablation studies in Appendix B for performance analysis of specific aspects of CAB. In the revised manuscript, we added blue text further highligting the most crucial components of the CAB. For a summary, from the empirical findings, the lagged cross-correlation filtering component (even when we hard-fix the learnable parameters as constants) resulted in highest increase in the accuracy of 1.39\% from the basic CAB model without lagged cross-correlation. Then, given that the lagged-cross correlation filtering is used, letting the parameter $\beta$, which is in Eq. (6) of our paper and balances the effect of instantaneous and lagged cross-correlation, be learnable would result in the highest increase in accuracy of 1.81\%.
>
>
>
> $\textbf{Q2:}$  "Could you elaborate on the primary challenges or obstacles preventing the integration of CAB into encoder-decoder models for conducting multivariate time series forecasting?";
> "It's important to acknowledge that the design of CAB is limited to encoder-only models and does not support time series forecasting. While this limitation is briefly mentioned, expanding on the reasons behind this constraint and discussing potential avenues for future work or extensions to address this limitation would add depth to the paper."
>
> $\textbf{Response:}$ Before providing detailed explanations, we would like to note that:
> -  We provide new experiments in Appendix C of the revised manuscript demonstrating that the CAB, even when naively plugged into the decoder architecture of base architectures, indeed consistently improves the performance of such Transformer-models over three long-term forecasting datasets spanning different domains!
> - We believe that the integration of CAB into the decoder can be done even better in the future with proper design of masking mechanism, as discussed in details below.
>
> First, we recall the main difference between the decoder and the encoder is that the decoder has two multi-head attention (MHA) blocks: one with masking and one without masking (i.e. conventional MHA). The masking is meant to prevent information to preserve the auto-regressive property, so that the forecasting of the next time step should depend only on the time steps strictly before it. In typical self-attention mechanism, this is done by  masking out (setting to $-\infty$) all values in the lower triangular of the scoring matrix $Q K^T\in \mathbb{R}^{T \times T}$ that is input to softmax. As you can see, $Q K^T$ can be thought of as the attention/correlation matrix in time (with dimension $T\times T$), so masking out the triangular part corresponds to "blinding" the history in making predictions; one the other hand, in CAB, the $Roll(K, l)^T Q \in \mathbb{R}^{d_k \times d_k}$, which is input to softmax, is correlation matrix across feature with dimension $d_k \times d_k$ instead of time, so the same masking strategy, even if naively applicable, would not have intuitive or physical phenomena as in vanilla self-attention, thereby being more of a heuristic approach. In the experiments in Appendix C, we only replace the non-masked MHA with our mixture-of-head attetions comprised of temporal attentions and our CAB, while keeping the masked MHA of the base Transformer. As you can see from the results, the CAB consistently improves performance of Non-stationary Transformer across all the tested datasets and resulting in the model that outperforms the previous SOTA Transformers for long-term forecasting.

---

> > ### Author Response · Authors · 2023-11-16
> > **Reply 1.2 to Reviewer M5Eq**
> >
> > $\textbf{Q3:}$ "Performance of CAB compared to vanilla CAB or previous baselines. Providing concrete numbers or performance metrics would strengthen the paper's findings in this regard."
> >
> > $\textbf{Response:}$ We concur with the reviewer and thus have provided new experiments on run-time analysis in Appendix D of the revised manuscript. The results show that the CAB incurs only minimal overhead to the base Transformers, so that the resulting base-Transformer+CAB's all approximately maintain the same order of computation magnitude as other SOTA Transformers. We note that those two baslines Timesnet and DLinear with superior computation overhead are non-Transformers and sub-optimal in performance.
> >
> >
> >
> >
> > $\textbf{Q4:}$ Typos and reference format
> >
> > $\textbf{Response:}$ We thank the reviewer very much for all the suggestions in presentation. We have fixed the typos into the revised manuscript, and will unify the reference format right in the next update. (As our reference list is actually quite long and of 3.5 pages, it will take some time for us to properly take care of all of the references.)
> >
> > $\textbf{Q5:}$ "In Table 2, it is evident that for the first three datasets, the TimesNet baseline consistently outperforms CAB when the mask ratio exceeds 25%. Could you provide insights into this performance discrepancy?"
> >
> > $\textbf{Response:}$ First, we note that the TimesNet baseline mostly outpeforms CAB on those ETTm1, ETTm2, and ETTh1 datasets. As these datasets do not have very large number of features (7 features), there can be less benefit from CAB. Second, unlike other task, imputation involves missing data, which may affect the learning of \emph{lagged} cross-correlation in view of the missing lags or time steps, so CAB may perform less effectively when the mask ratio is too large. Nevertheless, we would like to note that the  overall performance (averaged over all mask ratio) of Non-stationary+CAB is still better than TimesNet.
> >
> >
> >
> > ------
> >
> > We appreciate your insightful comments and again hope that you can positively re-evaluate our work if the concerns are all properly addressed. In the meantime, we would be happy to answer to any remaining concern of yours.

---

> > > ### Comment · Reviewer_M5Eq · 2023-11-22
> > > **Thanks for the response**
> > >
> > > Thanks for the response, and I'm willing to raise my score to 6 given most of my concerns are resolved.

---

> > > > ### Author Response · Authors · 2023-11-22
> > > > **Thank you note**
> > > >
> > > > Dear Reviewer M5Eq,
> > > >
> > > > We thank you very much for raising the score to 6 for our paper.
> > > > Please kindly note that our replies to your concerns on the discussion of the main components for CAB's efficiency, additional tests on computation and forecasting tasks have been all integrated into our revised manuscript. Your comments on the reference style will be done soon in our future revision. We appreciate all of your constructive reviews.
> > > >
> > > > Thank you again and best regards,
> > > >
> > > > Authors

---

> ### Author Response · Authors · 2023-11-19
> **Discussion Period**
>
> If you have further concerns regarding our answers, we are more than happy to respond to them during this discussion period. Otherwise, we would really appreciate your acknowledgement that you concerns indeed have been cleared.
>
> Again, we thank you very much for your time and reviews!

---

> ### Author Response · Authors · 2023-11-22
> **End of Discussion Period**
>
> Dear Reviewer,
>
> We kindly remind that the discussion period is coming to the end. Thus, please provide us with any concern you may still have. Otherwise, we would really appreciate your acknowledgement that you concerns indeed have been cleared and hope that you consider raising the score for our work, especially given the additional experiments having been added supporting our method.
>
> Again, we thank you very much for your time and and hard work reviewing our paper!
>
> Authors

---

> > ### Author Response · Authors · 2023-11-22
> > **Reminder till the end of discussion period**
> >
> > Dear Reviewer,
> >
> > This is a gentle reminder for the case of our paper, as it is  less than 24 hours till the end of discussion period.
> >
> > Please kindly note that Reviewer mjBo has very positively increased the score to acceptance threshold, after we provided new long-term forecasting results on the high-dimensional Traffic dataset; more details can be found in our response therein to him/her.
> >
> > Up till now, we have provided comprehensive results on 4 datasets for long-term forecasting showing the benefit and state-of-the-art Transformer results from CAB on 3 out of 4 tested datasets even on this task, and empirically shown that our CAB module incurs very minimal computational overhead (for both high-dimensional data and scaling input series length).
> >
> > Furthermore, we will publish the code, if the paper may get accepted. Thanks to the modularity of CAB, which can be seamlessly integrated into existing base Transformers, we believe that this will be beneficial to the community.
> >
> > If your concerns have been all properly clarified, we hope that you can positively re-evaluate our work up to the acceptance threshold.
> >
> > Thank you again for your time and review of the paper, which has helped us to improve our work.
> >
> > Authors

---

### Meta-Review · Area_Chair_5txf · 2023-12-06

**Metareview:**

This manuscript addresses an important challenge in multi-variate time-series analysis that requires modelling both the temporal and parameter correlations simultaneously (Reviewer M5Eq, mjBo, KQRi, rWhs). Results are impressive (Reviewer M5Eq, mjBo, KQRi).

Weakness:
Compared to the computational complexities, the introduced method seems to have very minor improvement. (Reviewer KQRi)

**Justification For Why Not Higher Score:**

Authors have not addressed  some of the reviewer comments objectively:
1. Although the CAB presents better accuracies, there are still some performance discrepancies with TimesNet,  (Reviewer M5Eq, Reviewer KQRi)
2. Although the performance of CAB is better is anomaly detection and classification, this was not observed in the imputation task. Furthermore, the computational complexity does not justify the improvement in accuracies. (Reviewer KQRi)

**Justification For Why Not Lower Score:**

N/A

This is an interesting work, esp in multi-variate time-series data analysis. I hope the constructive feedback from this review inspires the authors to edit the manuscript for more impact and publish in other venues.

---

### Decision · Program_Chairs · 2024-01-16

Reject